# *InfoGS*: Efficient Structure-Aware 3D Gaussians via Lightweight Information Shaping

**Yunchao Zhang**[1]   **Guandao Yang**[2]   **Leonidas Guibas**[2]   **Yanchao Yang**[1]

[1] The University of Hong Kong   [2] Stanford University

## ABSTRACT

3D Gaussians, as an explicit scene representation, typically involve thousands to millions of elements per scene. This makes it challenging to control the scene in ways that reflect the underlying semantics, where the number of independent entities is typically much smaller. *Especially,* if one wants to animate or edit objects in the scene, as this requires coordination among the many Gaussians involved in representing each object. To address this issue, we develop a mutual information shaping technique that enforces resonance and coordination between correlated Gaussians via a Gaussian attribute decoding network. Such correlations can be learned from putative 2D object masks in different views. By approximating the mutual information with the gradients concerning the network parameters, our method ensures consistency between scene elements and enables efficient scene editing by operating on network parameters rather than massive Gaussians. In *particular,* we develop an effective learning pipeline named *InfoGS* with lightweight optimization to shape the attribute decoding network, while ensuring that the shaping (consistency) is maintained during continuous edits, avoiding re-shaping after parameter changes. *Notably,* our training only touches a small fraction of all Gaussians in the scene yet attains the desired correlated behavior according to the underlying scene structure. The proposed technique is evaluated on challenging scenes and demonstrates significant performance improvements in 3D object segmentation and promoting scene interactions, while inducing low computation and memory requirements. Our code is available at: https://github.com/StylesZhang/InfoGS.

## 1 INTRODUCTION

Open-world 3D scene understanding and editing play important roles in gaming, AR/VR, and robotics applications. Recently, 3D Gaussian Splatting (3DGS) (Kerbl et al., 2023), has significantly advanced both the rendering quality and inference efficiency in 3D learning by representing scenes through a set of 3D Gaussians. Moreover, follow-up works in scene editing (Cen et al., 2023; Zhou et al., 2023; Ye et al., 2023) propose augmenting each Gaussian's parameters with task-relevant attributes, allowing users to edit scenes by selecting Gaussians based on these attributes and modifying their colors or positions. Despite the progress with 3DGS-based scene editing, the use of point-wise 3D Gaussians presents limitations, among which, the fact that the number of distinct and interactive entities in a scene is typically far fewer than the number of Gaussians is neglected. Furthermore, augmenting attributes of Gaussians introduce significant storage overhead and do not fundamentally address the lack of correlation between Gaussians.

The lack of intrinsic correlation among scene elements renders a clear contrast between the editing of 3D Gaussians and the perturbation in the real world. For example, real-world scenes consist of particles that aggregate into cohesive objects through physical or chemical bonds, forming highly organized, interdependent structures. Interactions with a small part of an object can induce changes throughout the entire entity due to these intrinsic correlations. Such a phenomenon implies a high degree of mutual information between particles of the same entity, and even across distinct objects. Ideally, 3D scene representations should capture these correlation structures, facilitating efficient learning and interaction for tasks like scene editing. If so, modifying the state of a single element could result in coherent adjustment in the states of all correlated elements, which promotes efficiency by eliminating the need to individually examine and edit each Gaussian.

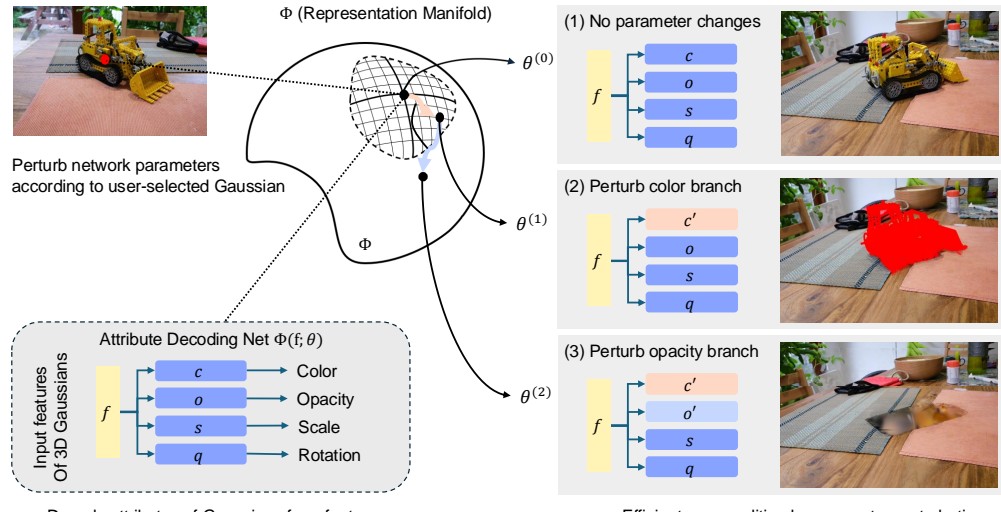

Figure 1: Our InfoGS is a mutual information shaping technique of the attribute decoding network based on 3D Gaussian Splatting (Kerbl et al., 2023). It can capture the underlying structure of the scene, while maintaining the correlations after consecutive parameter changes according to user-selected Gaussian (specified in Sec. 3.7). It promotes efficient scene editing by perturbing the network parameters, including re-colorization, segmentation, object removal, etc.

To enable efficient object-level interactive tasks, we propose the key is to enforce the inherent correlation structure within the scene representation, with which, the propagation of changes across correlated Gaussians can be easily achieved. Instead of assigning additional attributes to each Gaussian for grouping purposes, we introduce a novel learning scheme that encodes the correlations into the representation and an effective approach for manipulating scenes. Note that, a few 3DGS-based compression methods have exploited similarities between Gaussians to reduce storage costs, such as clustering Gaussians with similar parameter values (Fan et al., 2023), introducing anchors to group nearby Gaussians in different spatial regions (Lu et al., 2023), or organizing 3D space with feature closeness (Chen et al., 2024). These techniques primarily target scene reconstruction and rendering, however, we propose an explicit modeling of the correlations from the perspective of mutual information maximization to enable efficient scene editing.

We develop upon the common practice of 3DGS compression methods, where Gaussian attributes (used for reconstruction) are decoded from their features, and we introduce a mutual information shaping, named **InfoGS**, scheme to explicitly enforce the correlations between Gaussians via the feature-decoding network. This scheme ensures that perturbing the network parameters results in coherent changes among correlated Gaussians, enabling controllable and semantically meaningful interactions with objects in the rendered scene. Our strategy circumvents the need to estimate and encode pairwise feature correlations among millions of Gaussians. Instead, it constructs a well-organized tangent space that captures the underlying structure of the scene within the network.

Specifically, we devise a shaping loss that maximizes the mutual information (MI) between Gaussians belonging to the same entity (as evidenced by putative metrics), which correlates Gaussians associated with the same object and de-correlates otherwise. Furthermore, mutual information between Gaussians is approximated by the cosine similarity between the Jacobians with respect to the perturbed weights. This legitimates the tangent space of the feature-decoding network parameters to encode the desired correlation structure. However, simply shaping the tangent space *does not* guarantee that the perturbed network will preserve this correlation structure when the shaped weights are modified, i.e., the after-modification parameters may not have a shaped tangent space, thus, making it incapable of consistent perturbations across multiple editing operations. Another challenge lies in the computational and memory overhead required to calculate Jacobians, as this necessitates storing the full computational graph regarding the derivative of the Jacobians – a significant cost for 3DGS, where storage and training efficiency are critical concerns.

To further overcome these challenges, we propose shaping the activations of the perturbed weights instead of the Jacobians. We prove that this technique ensures that the network undergoes a conformal transformation in its tangent space, maintaining the correlation structure and enabling consistent editing across sequential operations. In other words, the proposed is a sufficient condition for mutual information maximization while inducing the benefits not possessed by directly shaping Jacobians. Finally, we develop an efficient pipeline that shapes the feature-decoding network with consistency preserved along a sequence of perturbations, while being computationally efficient. This pipeline is fully automated and light-weight. We evaluate the proposed MI shaping technique across various scene-editing applications, including 3D object removal, inpainting, colorization, and scene recomposition.

In summary, **1**) we propose a novel approach that leverages semantic correlations between 3D Gaussians for efficient scene editing; **2**) We design an efficient training pipeline that shapes the scene representation once and ensures that the perturbed representation continues to support coherent edits without reshaping along the perturbation trajectory; **3**) We demonstrate that our MI shaping technique affects only approximately $\sim 7\%$ of all Gaussians during training, while existing scene-editing methods require optimizing the entire set of Gaussians. Despite the lightweightness, our method achieves significant performance gains in tasks such as 3D segmentation, object removal, and inpainting, while substantially reducing computational and memory overhead.

## 2 RELATED WORKS

**Neural representations for 3D reconstruction**  Implicit neural representations have recently transformed the field of 3D reconstruction, such as Neural Radiance Fields (NeRF) (Mildenhall et al., 2021) and its extensions (Liu et al., 2020; Karras et al., 2021; Martin-Brualla et al., 2021; Barron et al., 2022; Deng et al., 2022; Zhang et al., 2023), which utilize neural networks to parameterize 3D scenes, enabling detailed and accurate reconstructions. Subsequent advancements have focused on optimizing the NeRF framework for large-scale scenes. Techniques such as multi-resolution hash tables and voxel grids have been proposed to compress scene representations (Garbin et al., 2021; Reiser et al., 2021; Müller et al., 2022). Additionally, hybrid approaches have incorporated explicit modeling strategies, leveraging point clouds to enhance scene representation (Fridovich-Keil et al., 2022; Xu et al., 2022; Kerbl et al., 2023). Notably, 3D Gaussian Splatting (3DGS) (Kerbl et al., 2023) employs point-wise Gaussians to achieve efficient training and rendering. These 3D representations are useful for various downstream tasks, such as semantic understanding (Zhi et al., 2021; Gao et al., 2022; Bao et al., 2023; Wang et al., 2023b; Li et al., 2024), segmentation (Fu et al., 2022; Ye et al., 2023; Cen et al., 2023), and 3D content creation (Tang et al., 2023; Chen et al., 2023b). Nevertheless, current methods predominantly rely on first-order supervision, often neglecting the higher-order correlations encoded in the scene representation.

**Scene representations for 3D editing**  Despite advancements in reconstruction, interactive 3D scene editing remains challenging. Current neural representation-based approaches (Schwarz et al., 2020; Wang et al., 2022; Yuan et al., 2022; Wang et al., 2023a; Zhou et al., 2023; Chen et al., 2023a) typically rely on attributes computed by distilling features from 2D foundation models and perform editing in a point-wise manner. In contrast, our work delves into manipulating scenes based on a second-order correlation structure. The idea of enforcing second-order correlation is first proposed in JacobiNeRF (Xu et al., 2023), while it mainly focuses on label propagation and one-time perturbation consistency, limiting its potential for diverse scene manipulations. Furthermore, the correlation shaping in Xu et al. (2023) is both memory- and time-intensive during training. Later, InfoNorm (Wang et al., 2024) has shown that enforcing second-order correlation also facilitates geometric reconstruction from a few views. Other approaches involve using physics simulators to guide scene editing (Li et al., 2023; Xie et al., 2023), yet incurring a high computational cost in solving partial differential equations at inference time. Our approach offers efficient optimization and enables fast scene editing during inference by learning continuously well-shaped tangent spaces.

## 3 INFOGS: MODELING MUTUAL INFORMATION BETWEEN 3D GAUSSIANS

Our goal is to develop scene representations that not only capture 3D geometry and appearance with high precision but also encode the interrelationships between individual elements. These enriched

representations enable efficient scene editing by ensuring coherent and realistic updates, where changes propagate in a coordinated and intuitive manner.

Building on 3D Gaussian Splatting (3DGS) (Kerbl et al., 2023), a state-of-the-art technique for 3D reconstruction that represents scenes using independent sets of 3D Gaussians, we aim to introduce meaningful correlations between these Gaussians. To achieve this, we incorporate mutual information (MI) to capture their underlying interdependence. However, traditional MI-based methods involve heavy optimization processes and are typically applied point-wise, meaning they are valid only for the current representation. This renders them unsuitable for scene editing, which requires both real-time response and persistent editability.

To overcome this limitation, we propose a novel training approach that efficiently shapes the attribute decoding network to capture and maintain these correlations throughout the decoding process. Notably, these structured correlations persist even after multiple consecutive scene edits. Finally, we demonstrate how these well-formed correlations improve a range of downstream scene editing tasks.

## 3.1 Preliminaries: 3D Gaussian splatting and attibute decoding

Given a dataset $\mathcal{D}$ consisting of multi-view 2D images and corresponding camera poses, 3D Gaussian Splatting (3DGS) reconstructs the 3D scene by learning a set of 3D Gaussians $\mathcal{G} = \{\mathbf{g}_1, \mathbf{g}_2, ..., \mathbf{g}_N\}$, where $N$ denotes the number of 3D Gaussians in the entire scene. Each 3D Gaussian $\mathbf{g}_i$ possesses multiple attributes, denoted as $\{\mathbf{x}_i, \mathbf{s}_i, \mathbf{q}_i, \mathbf{o}_i, \mathbf{c}_i\}$. Specifically, $\mathbf{x}_i \in \mathbb{R}^3$ represents the location of the centroid of $\mathbf{g}_i$, $\mathbf{s}_i \in \mathbb{R}^3$ denotes the scale, and $\mathbf{q}_i \in \mathbb{R}^4$ denotes the rotational quaternion, such that $\mathbf{s}_i$ and $\mathbf{q}_i$ jointly represent the 3D covariance matrix $\boldsymbol{\Sigma}$ of $\mathbf{g}_i$. Additionally, $\mathbf{o}_i \in \mathbb{R}$ represents the opacity, and $\mathbf{c}_i$ denotes the color, which is represented as the spherical harmonics (SH) coefficients. Given a camera pose, the color $C(p)$ of a pixel $p$ can be computed by blending a set of depth-ordered Gaussians $\mathcal{N}$ that overlap with $p$:

$$C(p) = \sum_{i \in \mathcal{N}} \mathbf{c}i\alpha_i \prod_{j=1}^{i-1}(1 - \alpha_j), \tag{1}$$

where $\mathbf{c}_i$ is the color of the $i$-th Gaussian, and $\alpha_i$ is the blending weight, calculated using the opacity, projected 2D covariance of the Gaussian, and the location of the pixel.

While 3DGS offers several advantages, its sparse and unorganized structure leads to significant storage demands and makes scalability challenging. To address this, mutual relationships between Gaussians have been explored in subsequent works to eliminate structural redundancies. Lu et al. (2023); Lee et al. (2024) proposed leveraging spatial relationships (i.e., features of Gaussians in a local region tend to have similar values) by employing neural networks to decode their features:

$$a(\mathbf{g}_i) = \boldsymbol{\Phi}_a(f(\mathbf{g}_i), d; \theta), \tag{2}$$

where $a(\mathbf{g}_i)$ denotes attributes such as the color $\mathbf{c}_i$ of the Gaussian $\mathbf{g}_i$. The Gaussian attributes are decoded by an attribute decoding network $\boldsymbol{\Phi}_a$ with parameters $\theta$, taking the feature of the Gaussian $f(\mathbf{g}_i)$ and the viewing direction $d$ as inputs. Incorporating the viewing direction into inputs improves the scene representation's robustness to substantial viewpoint changes and lighting effects.

Although associating millions of Gaussians via a network has proven effective, we argue that deeper exploration of the correlation structure of Gaussians through mutual information can further enhance scene editing, enabling better coordination among Gaussians for more effective results.

## 3.2 Mutual information between 3D gaussians

Traditional 3DGS losses predominantly focus on first-order supervision, optimizing for view synthesis rather than scene interaction. In contrast, this work introduces second-order supervision to shape the attribute decoding multilayer perceptron (MLP), facilitating efficient scene editing. Specifically, we optimize the network parameters to induce a global structure that encodes correlations among individual Gaussians. This ensures that modifications to the attribute decoding network's parameters, denoted by $\theta$, directly correspond to meaningful scene-editing operations. As a result, network outputs produce coherent and semantically consistent editing effects in the rendered scene, driven by the learned co-variation correlations.

To formalize this, we study the correlations between Gaussians under representation parameter changes in the output attributes. We analyze this behavior by considering two randomly selected Gaussians, $\mathbf{g}_i$ and $\mathbf{g}_j$, and their corresponding offsets in output attributes, $\hat{a}(\mathbf{g}_i)$ and $\hat{a}(\mathbf{g}_j)$. These attribute changes arise from the perturbed network $\hat{F}_a(\cdot; \theta^D + \mathbf{n})$, where $\mathbf{n} \in \mathbb{R}^D$ represents a perturbation applied to the parameter set $\theta^D$.

Following Xu et al. (2023), we show that the mutual information between the attribute variations $\hat{a}(\mathbf{g}_i)$ and $\hat{a}(\mathbf{g}_j)$ is proportional to the cosine similarity of the Jacobians of $\boldsymbol{\Phi}_a$ with respect to the perturbed parameters $\theta_D$:

$$\mathbb{I}(\hat{a}(\mathbf{g}_i), \hat{a}(\mathbf{g}_j)) = \log\left(\frac{1}{\sqrt{1 - \cos^2(\gamma)}}\right) + \text{const.}, \tag{3}$$

where $\gamma$ denotes the angle between the Jacobians $\frac{\partial \boldsymbol{\Phi}_a(f(\mathbf{g}_i,d);\theta)}{\partial \theta_D}$ and $\frac{\partial \boldsymbol{\Phi}_a(f(\mathbf{g}_j),d;\theta)}{\partial \theta_D}$. For clarity, we denote the Jacobians as $\partial \boldsymbol{\Phi}_i$ and $\partial \boldsymbol{\Phi}_j$. This formulation reveals that the mutual information $\mathbb{I}(\hat{a}(\mathbf{g}_i), \hat{a}(\mathbf{g}_j))$ is positively correlated with the cosine similarity of their gradients with respect to the perturbed parameters.

We apply Eq. 3 to evaluate whether the correlations between Gaussians are well-structured by the attribute decoding network $\boldsymbol{\Phi}_a$. As shown in Fig. 2 (a), network weights optimized solely by the reconstruction loss fail to reveal meaningful correlations between Gaussians, primarily due to the unstructured nature of 3DGS. Despite the association of millions of Gaussians via the decoding network, they remain disorganized— a limitation of methods optimized solely for novel view synthesis.

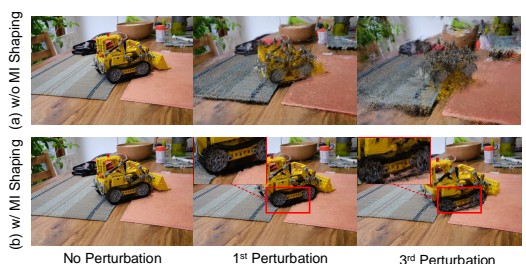

Figure 2: Perturb the attribute decoding network by the Jacobian of a Gaussian in the bulldozer without (a) or with (b) MI shaping (Xu et al., 2023), and then move Gaussians according to similarities of Jacobian with selected one.

JacobiNeRF (Xu et al., 2023) addresses this by enforcing perturbation consistency through maximizing the quantity in Eq. 3, enabling label propagation tasks. However, this approach is not suitable for scene editing in the 3DGS framework for two main reasons: (1) It does not support consecutive scene editing. As shown in Fig. 2 (b), previous MI shaping fails after multiple editing operations, leading to unrealistic scene updates. This is because the shaping in Xu et al. (2023) is *point-wise*: while $\partial \boldsymbol{\Phi}_i$ and $\partial \boldsymbol{\Phi}_j$ may be similar in the original tangent space, there is no guarantee that they remain similar after parameter perturbation. (2) The optimization process is complex, requiring substantial storage to maintain the computational graph and significant time for Jacobian computation. This approach is inefficient for 3DGS and fails to meet the real-time demands of user-driven scene interactions.

Next, we will discuss how to ensure consistency across consecutive parameter changes while maintaining training efficiency.

### 3.3 MUTUAL INFORMATION SHAPING WITH CONSISTENCY AND EFFICIENCY

To ensure consistency in correlation shaping after perturbation, we begin by formulating the gradient expression for an in-depth analysis. Given that attribute decoding networks in neural fields (e.g., NeRFs and 3DGS) are MLPs, we focus on perturbing the weights $W^{(l)}$ of the $l$-th linear layer in the network $\boldsymbol{\Phi}_a$ (i.e., $\theta_D = W^{(l)}$), expressed as $h^{(l)} = W^{(l)}\sigma(h^{(l-1)}) + b^{(l)}$. Here, $\sigma$ denotes the non-linear activation function, $b^{(l)}$ is the bias, and $h^{(l)}$ is the (hidden) output of the $l$-th layer.

To reveal the relationship between the gradient of a Gaussian $\mathbf{g}_i$, $\partial \boldsymbol{\Phi}_i = \frac{\partial \boldsymbol{\Phi}(f(\mathbf{g}_i,d);\theta)}{\partial \theta_D}$, and the perturbed weight $W^{(l)}$, we consider the gradient of an arbitrary scalar output $z$ in the attribute decoding network with respect to $W^{(l)}$:

$$z = Y(h^{(l)}) = Y(W^{(l)}\sigma(h^{(l-1)}) + b^{(l)}), \tag{4}$$

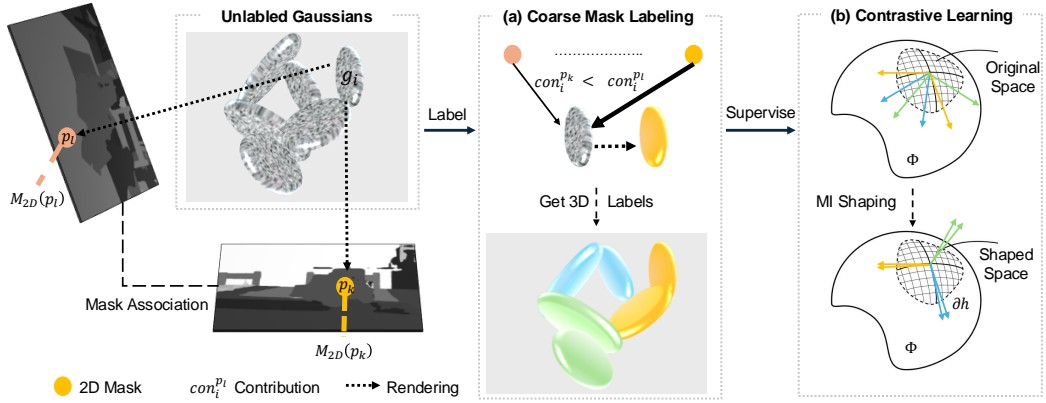

Figure 3: The training pipeline of InfoGS: **(a)** We use SAM (Kirillov et al., 2023) to generate 2D masks and a pre-trained zero-shot tracker (Cheng et al., 2023) to associate masks from different views. We label the 3D mask of each Gaussian according to the 2D mask of the pixel that owns the Gaussian's maximal contribution during rendering across all views. **(b)** We use the labeled 3D masks as the supervision to conduct contrastive learning for mutual information shaping. After shaping, the Jacobians are consistently distributed in the tangent space.

where $Y$ denotes all transformations after the $l$-th layer, and $z$ is an arbitrary scalar in the vector output of $\Phi$. The differential $\mathrm{d}z$ can be written as:

$$\mathrm{d}z = \frac{\partial z}{\partial h^{(l)}}^{\top} \mathrm{d}h^{(l)} = \sigma(h^{(l-1)}) \frac{\partial z}{\partial h^{(l)}}^{\top} \mathrm{d}W^{(l)} = \mathrm{tr}(\sigma(h^{(l-1)}) \frac{\partial z}{\partial h^{(l)}}^{\top} \mathrm{d}W^{(l)}), \qquad (5)$$

since $\mathrm{d}z = \mathrm{tr}\left(\frac{\partial z}{\partial W^{(l)}}^{\top} \mathrm{d}W^{(l)}\right)$, we can derive:

$$\frac{\partial z}{\partial W^{(l)}} = \frac{\partial z}{\partial h^{(l)}} \sigma(h^{(l-1)})^{\top}. \qquad (6)$$

By expressing $\cos(\gamma)$ in Eq. 3 using the right-hand side of Eq. 6, we derive:

$$\cos(\partial \Phi_i^{(d)}, \partial \Phi_j^{(d)}) \approx \cos(\partial h_i^{(0)}, \partial h_j^{(0)}), \quad d \in \mathbb{N}, \qquad (7)$$

when $\partial h_i^{(0)}$ and $\partial h_j^{(0)}$ point in the same or opposite directions. Here, $\partial \Phi_i^{(d)}$ represents the Jacobian after the $d$-th perturbation, and $\partial h_i^{(0)}$ represents the Jacobian $\frac{\partial h^{(l)}}{\partial W^{(l)}}$ of $\mathbf{g}_i$ without perturbation. Upon further derivation, we find that $\partial h$ corresponds to repeated activations $\sigma(h^{(l-1)})$. A detailed derivation and the proof of Eq. 7 can be found in the Appendix C. Eq. 7 indicates that activation shaping can substitute for Jacobian calculation in Eq. 7 to satisfy object-level mutual relations, i.e., $\|\cos(\gamma)\|$ approaches 1 if $\mathbf{g}_i$ and $\mathbf{g}_j$ belong to the same object, or approaches 0 if not.

The key insight of Eq. 7 is that by optimizing activations instead of Jacobians, the correlations between Gaussians can be correctly shaped and remain consistent after successive parameter changes. This enables versatile editing operations in the scene and makes training much more efficient, as there is no need to maintain the computational graph for Jacobian calculation.

## 3.4 SHAPING CORRELATIONS BETWEEN GAUSSIANS FOR SCENE EDITING

According to Eq. 3 and Eq. 7, to ensure that two 3D Gaussians remain highly correlated consistently even after parameter changes – thus supporting a sequence of editing operations without reshaping the parameters – we shape their activations $\partial h$ within the attribute decoding network $\Phi a$. To enable meaningful scene editing, we define positive and negative pairs based on object-level relationships, where $(g_i, g_i^+)$ represents a pair of Gaussians belonging to the same entity, and $(g_i, g_i^-)$ are independent Gaussians. For positive pairs, we align their activations in $\Phi a$ relative to the perturbed layer $W^{(l)}$, while for negative pairs with minimal mutual information, we enforce their $\partial h$ to be orthogonal. To achieve this, we minimize the InfoNCE loss (van den Oord et al., 2018):

$$\mathcal{L}_{\mathrm{MI}} = -\log \frac{\exp(|\cos(\partial h_i, \partial h_{i+})|/\tau)}{\sum_{i^+ \cup \{i^-\}} \exp(|\cos(\partial h_i, \partial h_{i-})|/\tau)}, \qquad (8)$$

where $\tau$ is the temperature parameter. This loss encourages highly correlated Gaussians to exhibit large activation similarity while ensuring that uncorrelated Gaussians exhibit near-zero similarity.

## 3.5 SUPERVISION REFINEMENT

We now address how to sample positive and negative pairs from all 3D Gaussians. Identifying the label of each 3D Gaussian is challenging, as 3DGS is trained from a set of 2D posed images, complicating the recovery of the joint 3D distribution. However, by leveraging the advances of 2D vision foundation models (Kirillov et al., 2023), we label a coarse mask by lifting 2D mask to 3D:

$$M_{3D}(g_i) = M_{2D}(\arg\max_p con_i^p),\tag{9}$$

where $con_i^p$ represents the contribution of $g_i$ to pixel $p$ during rendering, and $M_{3D}$ and $M_{2D}$ denote the 3D mask of Gaussians and the 2D mask of pixels. Eq. 9 indicates that the mask of a 3D Gaussian $g_i$ is determined by the 2D mask of a pixel where $g_i$ contributes the most across all 2D images. We compute the contribution $con_i^p$ based on a trained 3DGS that contains geometric priors of the scene. Thus the mask of $g_i$ is likely to be determined by the 2D mask where its maximal contribution appears. In order to harmonize 2D mask IDs produced from different views, we use a pre-trained zero-shot tracker (Cheng et al., 2023) to associate 2D masks by treating them as a video sequence. We also provide a version without using a tracker in the experiments, and it is elaborated in the Appendix A.

To further enhance efficiency, we introduce a smoothness regularization in the self-supervised training. Since objects are continuous in space, neighboring Gaussians of $\mathbf{g}i$ are likely to belong to the same object as $\mathbf{g}_i$. This observation helps shape the Jacobians of 3D Gaussians that are heavily occluded by others and may not be accurately annotated in the coarse 3D segmentation. Such Gaussians can be supervised by nearby, well-labeled Gaussians. Specifically, $\mathcal{L}_R$ is formulated as:

$$\mathcal{L}_R = \mathbb{E}_{\mathbb{P}}[\frac{1}{k}\sum_{x^+ \in n(x)} 1.0 - \cos(\boldsymbol{\Phi}_x, \boldsymbol{\Phi}_{x^+})],\tag{10}$$

where $\mathbb{P}$ denotes the input space, $n(x)$ represents the neighboring Gaussians of $x$, and $k = \|n(x)\|$ is the number of neighbors. In practice, the distance between two Gaussians is defined by the Euclidean distance between their centroids $\mathbf{x}$, and $k$ is a fixed hyperparameter during training.

## 3.6 FULL PIPELINE OF INFOGS

Fig. 3 illustrates our training pipeline of InfoGS for the correlation shaping, and the loss is as follows:

$$\mathcal{L} = \mathcal{L}_{3DGS} + \lambda_m\mathcal{L}_{MI} + \lambda_r\mathcal{L}_R,\tag{11}$$

where $\mathcal{L}_{3DGS}$ is the photometric reconstruction loss from the original 3DGS pipeline, $\mathcal{L}_{MI}$ is the contrastive learning loss (Eq. 8), and $\mathcal{L}_R$ is the regularization loss, with $\lambda_r$ representing its corresponding weight.

## 3.7 CORRELATION-BASED SCENE EDITING

After shaping the correlations, scene editing is performed by perturbing the parameters of the attribute decoding network. Specifically, we perturb the attribute decoding network by applying the average Jacobian $\partial\boldsymbol{\Phi}_a$ of user-selected Gaussians. When editing through a 2D image rather than directly in 3D space, we select the Gaussians that have the largest influence on the color of the user-selected pixels, similar to the contribution calculation in Eq. 9.

Different branches of the network produce distinct editing effects. For object removal, we perturb the opacity branch $\boldsymbol{\Phi}_o$ in the gradient direction that reduces output opacity. This can drive the opacity of the relevant Gaussians to zero, effectively removing the object in the rendered scene. For object re-colorization, we perturb the color branch $\boldsymbol{\Phi}_c$ along different gradient directions of output channels to generate diverse color changes. In the case of 3D segmentation and object movement, we move Gaussians according to similarities in Jacobian with the selected one. When extending to dynamic scenes (Pumarola et al., 2020), we achieve object movement by shaping and perturbing the motion network. Additional editing tasks are discussed in the Appendix B.

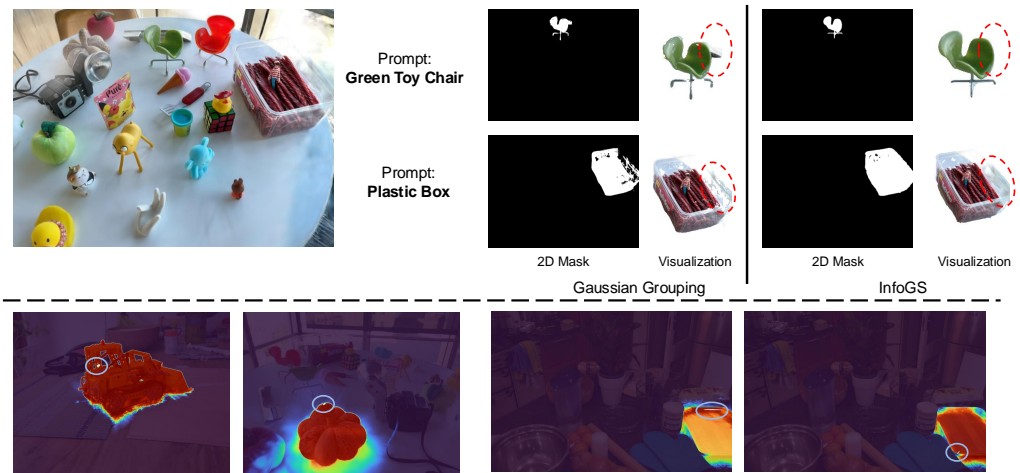

Figure 4: Top: qualitative results of open vocabulary segmentation (Ye et al., 2023). Bottom: gallery of relevance maps when perturbing a single Gaussian (highlighted in the purple circle).

## 4 EXPERIMENTS

### 4.1 IMPLEMENTATION DETAILS

**Training.** We shape mutual information between Gaussians starting from a pre-trained 3D Gaussian Splatting, applying a fine-tuning stage for optimizing attribute decoding network $\Phi_a$. We follow the shaping pipeline in Fig. 3 for 1500 iterations. The finetuning stage is conducted on a single RTX 3090 GPU for about 1 minute. Detailed hyper-parameter settings can be found in the Appendix A.

**Datasets.** We test InfoGS on open-world datasets LERF-Localization (Kerr et al., 2023), derived LERF-Mask dataset (Ye et al., 2023) with ground-truth segmentation, and Mip-NeRF 360 (Barron et al., 2022), in order to evaluate the segmentation and editing quality in complex and compositional scenarios. We also test the effectiveness of our shaping on the real dynamic scenes in the D-NeRF dataset (Pumarola et al., 2021). Additionally, we provide results on the outdoor unbounded scenes in dataset NERDS 360 (Irshad et al., 2023) in the Appendix B.

### 4.2 3D SEGMENTATION

After shaping the correlations among Gaussians, our approach enables 3D segmentation by simply perturbing the network parameters. We use Grounding DINO (Liu et al., 2023) to generate the 2D mask prompt. We compare InfoGS with the state-of-the-art open vocabulary 3D segmentation methods, including both the NeRF-based (Kerr et al., 2023; Cen et al., 2023), and 3DGS-based methods (Xu et al., 2023; Ye et al., 2023). It's worth noting that Gaussian Grouping (Ye et al., 2023) use the consistent 2D masks generated from the video tracker. To evaluate the effectiveness of our improvement on correlation shaping, we utilize the MI shaping loss in JacobiNeRF (Xu et al., 2023) as a substitute for our MI shaping $\mathcal{L}_{MI}$ in Eq. 11, which only permits one-time perturbation consistency. While keeping other loss terms and supervision refinement the same for a fair comparison, we name the baseline *JaocbiGS*. We also include the results of using foundation models.

For quantitative results, Tab. 1 demonstrates that ours significantly outperforms both NeRF-based and GS-based methods. Compared with previous state-of-the-art results, InfoGS increases the mIoU by averagely 11%. Qualitative results are shown in Fig. 4. We can see that InfoGS produces sharp boundaries for the segmentation without fewer blurs. We also provide the relevance map when perturbing a single Gaussian to select the object. The relevance scores are represented as the cosine similarities of the Jacobians $\partial\Phi$ of the perturbed Gaussian and all other Gaussians. As in Fig. 4, when perturbing a single Gaussian, the corresponding object is highlighted, evidencing ours capability of segmentation from another viewpoint. It also shows that the relevance of Jacobian would be higher if a Gaussian is closer to the perturbed one. These phenomena comply with our expectations, indicating the effectiveness of the proposed MI shaping.

Table 1: Quantitative results of Open Vocabulary Segmentation on LERF-Mask dataset. The time and memory costs mentioned refer to the total consumption of training and finetuning (if have). Comparisons with non-3DGS methods in this table are not feasible.

| Model | figurines | | ramen | | teatime | | Average Training Cost | | |
|---|---|---|---|---|---|---|---|---|---|
| | mIoU | mBIoU | mIoU | mBIoU | mIoU | mBIoU | Time/Minutes | Memory/M | #GS Used |
| LERF (Kerr et al., 2023) | 33.5 | 30.6 | 28.3 | 14.7 | 49.7 | 42.6 | * | * | * |
| DEVA (Cheng et al., 2023) | 46.2 | 45.1 | 56.8 | 51.1 | 54.3 | 52.2 | * | * | * |
| SA3D (Cen et al., 2023) | 24.9 | 23.8 | 7.4 | 7.0 | 42.5 | 39.2 | * | * | * |
| JacobiGS Xu et al. (2023) | 62.2 | 60.1 | 64.3 | 57.7 | 69.7 | 70.1 | 30.2 | 160.2 | 17% |
| Gaussian Grouping Ye et al. (2023) | 69.7 | 67.9 | 77.0 | 68.7 | 71.7 | 66.1 | 55.2 | 757.2 | 100% |
| InfoGS w/o DEVA | 72.5 | 70.3 | 79.1 | 69.9 | 77.8 | 70.2 | 19.7 | 160.2 | 15% |
| InfoGS | **80.6** | **77.2** | **80.6** | **70.1** | **84.5** | **78.7** | **16.3** | **160.2** | **7%** |

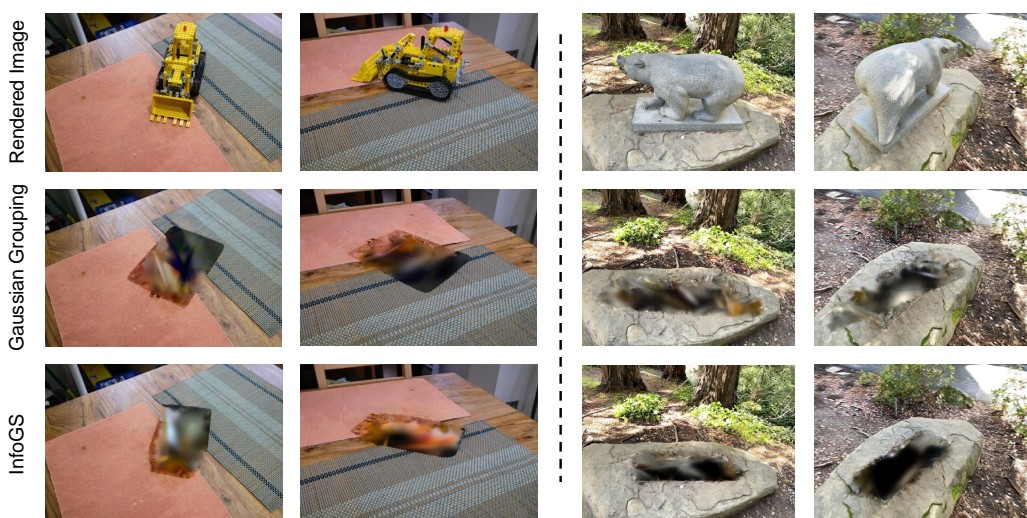

Figure 5: 3D object removal on bear & kitchen scenes. Compared to Gaussian Grouping, InfoGS removes the object with a more fitting curve and less distortion of the irrelevant area.

The superior performance of InfoGS can be attributed to the learned awareness of the scene structure. Instead of directly assigning scene attributes to each Gaussian (Cen et al., 2023; Zhou et al., 2023; Ye et al., 2023), we focus on shaping networks that act on the Gaussians in a way respecting the underlying structures. It helps a lot for low-level representation like 3DGS when leveraged for interactive tasks like segmentation and editting. The shaped MLP is aware of the entire scene structure rather than individual Gaussians. Notably, we only sample about 7% of all the Gaussians during finetuning until convergence, boosting the training time while with little memory storage.

### 4.3 SCENE EDITING

**3D Object removal.** 3D object removal is to delete the target object without affecting the background or unrelated area. The space behind the target is unknown from training views and thus it would be noisy after deletion. According to Fig. 5, InfoGS preserves more details of the unrelated area, and the contour of the area caused by the deletion fits the target object shape better. More explicitly, the noisy area resulted from InfoGS after removal in the bear scene accurately matches the rectangle shape of the pedestal of the bear statue. While both are supervised by SAM and DEVA, InfoGS separates the large object more clearly compared to Gaussian Grouping.

**3D Object movement.** We adopt the task of object movement to evaluate the editing quality under consecutive perturbations. A single perturbation is to add the multiplication of the Jacobian $\partial \Phi_i$ of selected Gaussian $g_i$ and a scaling factor $\sigma_s$ for controlling the impact. As shown in Fig. 6, perturbing scenes with the same Gaussian reveals that InfoGS induces reasonable motion (e.g., the carton is lifted up) while preserving the uncorrelated objects. It exhibits path-consistent motion after consecutive perturbations. In contrast, scenes generated by JacobiGS diverge into unrealistic representations after the second and third perturbation. For example, consecutive perturbations to the

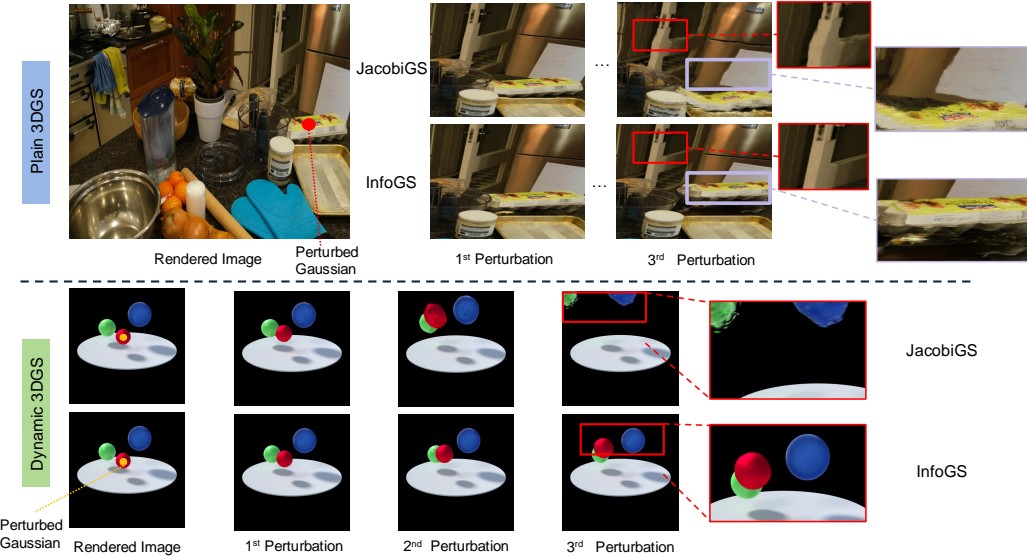

Figure 6: 3D object movement on Mip-NeRF 360 (Barron et al., 2022) and D-NeRF (Pumarola et al., 2020). Compared to the correlation shaping in Xu et al. (2023), InfoGS enables consecutive editing (no need for reshaping) on both static and dynamic scenes, while maintaining consistency.

Table 2: Ablation study of correlation shaping.

| Model | Mip-NeRF 360 | | | D-NeRF | | |
|---|---|---|---|---|---|---|
| | PSNR↑ | SSIM↑ | LPIPS↓ | PSNR↑ | SSIM↑ | LPIPS↓ |
| GS Baseline (Kerbl et al., 2023; Yang et al., 2023) | 28.69 | 0.870 | 0.182 | 41.0 | 0.995 | 0.009 |
| InfoGS | 28.72 | 0.842 | 0.208 | 39.7 | 0.993 | 0.012 |

carton result in the distortion of the door behind the box (see the red boxes for highlights in Fig. 6). The observations demonstrate that InfoGS supports continuous editing with consistency, thereby verifying the effectiveness of the proposed shaping technique.

### 4.4 THE EFFECT ON THE RECONSTRUCTION QUALITY

We analyze how correlation shaping affects reconstruction quality. From Tab. 2, we can see that our framework has comparable performance to the baseline method on both static and dynamic scenes. There is a small PSNR drop on dynamics scenes, as we simply integrate our shaping loss into the original deformation network in Yang et al. (2023), which aims for 4D reconstruction, making it a bit challenging to shape all timesteps, and this is subject to future study.

## 5 DISCUSSION

We attempt the challenge of shaping correlations between independent scene elements in a 3D Gaussian Scene (3DGS) model to enable scene editing through network parameter perturbations. Our derivation demonstrates that enforcing gradient alignment within the network ensures the consistency of mutual information between Gaussians, even after successive parameter perturbations. To achieve this, we introduce an efficient training pipeline, incorporating a contrastive loss and a 2D-to-3D sampling strategy. With the well-shaped correlation structure, we illustrate how the resonance between Gaussians facilitates 3D segmentation and automatically directs millions of Gaussians to respond appropriately to accomplish diverse editing tasks. Despite this efficiency, our current pipeline has limitations when generating arbitrarily complex edits, particularly at a fine-grained level, such as object deformation. In future work, we aim to address this limitation by learning intrinsic object features and kinematics from large-scale video datasets, which would enhance its capabilities in more sophisticated scene manipulation tasks.

## 6 ACKNOWLEDGMENT

This work is supported by the Early Career Scheme of the Research Grants Council (grant # 27207224), the HKU-100 Award, a donation from the Musketeers Foundation, the HKU Seed Fund for PI Research, and in part by the JC STEM Lab of Robotics for Soft Materials funded by The Hong Kong Jockey Club Charities Trust.

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

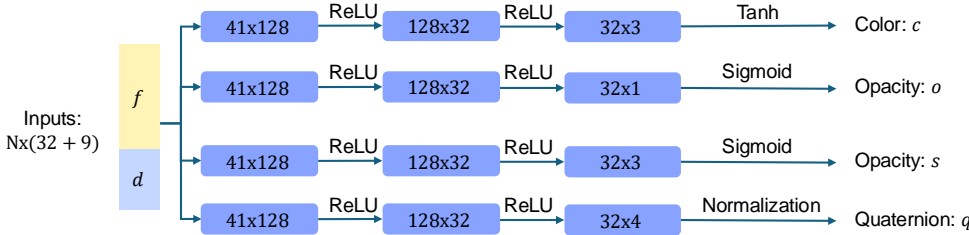

Figure 7: The structure of attribute decoding network. For each 3D Gaussian, we use different branches to predict attributes (opacity, color, scale, and quaternion). The inputs are features of Gaussians with view direction.

## A IMPLEMENTATION DETAILS

**Attribute decoding network.** The attribute decoding network is composed of four different branches, including the color branch $\mathbf{\Phi}_c$, opacity branch $\mathbf{\Phi}_o$, and branches for the covariance $\mathbf{\Phi}_s$ and $\mathbf{\Phi}_q$. They are all implemented by MLPs with RELU activation. The detailed structure is illustrated in Fig. 7, where the output activation of different branches is inspired from Lu et al. (2023). The centroid $x$ of each Gaussian is initialized by the pre-computed point cloud from SfM points (Özyeşil et al., 2017) and separately optimized. The hidden dimension of each feature is 32 and initialized to be zero.

**Training hyperparamters.** During training, we set the hyperparameters $\lambda_{MI} = 0.1$, $\lambda_R = 0.1$ and $k = 5$. We adopt the Adam optimizer Kingma & Ba (2014) when shaping the attribute decoding network, with a learning rate of $0.01$. The gradient that we use to shape the motion network is $\partial h = \frac{\partial h^{(l)}}{\partial W^{(l)}}$ of each branch by default, where $l$ is the linear layer before the final output layer. We train the network for $1500$ iterations, sampling a random view and a batch of $512$ 3D Gaussians at each iteration to compute the training loss.

When training in the style of previous mutual information shaping in JacobiNeRF (Xu et al., 2023) to produce the results of JacobiGS, we modify the sampling batch size from $512$ to $48$ due to the large memory consumption. This is because the shaping method in JacobiNeRF requires the direct calculation of Jacobians $\partial\mathbf{\Phi}$, which necessitates the preservation of the entire computation graph.

**Training without using the tracker for mask association.** We use DEVA (Cheng et al., 2023) to associate the masks produced from SAM (Kirillov et al., 2023) for training. However, the assumption of using the tracker to associate masks from different views is that the multi-view training images could be treated as a video. It will not hold when training views are sparse and random. In light of this, we introduce another shaping version without mask association, which is supervised by projecting the Jacobians to 2D for a sampled view:

$$\partial\mathbf{\Phi}(p) = \sum_{i\in\mathcal{N}} \partial\mathbf{\Phi}_i\alpha_i \prod_{j=1}^{i-1}(1-\alpha_j), \tag{12}$$

where $\partial\mathbf{\Phi}_i$ is the Jacobian of the $i$-th Gaussian and $\alpha_i$ is the blending weight that is calculated with the opacity, projected 2D covariance of the Gaussian, and the location of the pixel. By projecting the Jacobians of 3D Gaussians to 2D space, we can now sample Jacobians of pixels instead of 3D Gaussians for contrastive learning. We sample a random view and $512$ pixels of it for each iteration, supervised by the 2D mask of this view which is generated from SAM. We keep all other hyperparameters the same as the version of mask association. Note that we still use $\partial h$ to replace $\partial\mathbf{\Phi}$ for Jacobian shaping according to Eq. 7.

**Speed-up module.** For the MI shaping version without mask association, it would be time-consuming when projecting Jacobians to the 2D plane as the dimension of the Jacobian is high. We then propose a speed-up module that modifies the rasterization process in the CUDA pipeline. As we only need to sample a small batch of Jacobians of pixels at each iteration, we add a mask during the CUDA forward and backward process to avoid the calculation of non-sampled pixels for Jacobians projection.

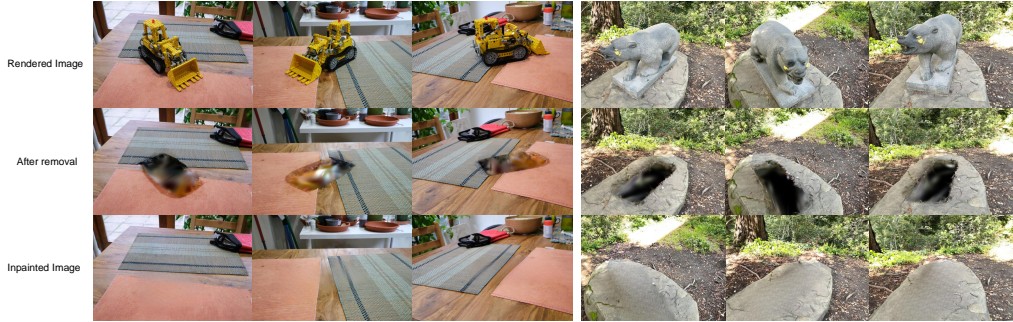

Figure 8: 3D object Inpainting of our method. We use LAMA (Suvorov et al., 2022) to inpaint the images after object removal, and then finetune Gaussians based on the inpainted 2D images.

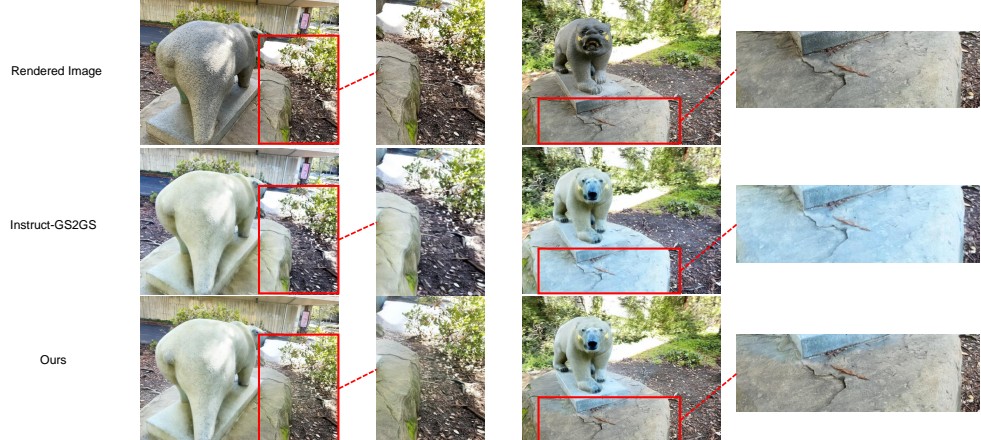

Figure 9: 3D object style transfer. We use Instruct-Pix2Pix (Brooks et al., 2023) to edit the 2D images (prompt: *"Turn the bear into a polar bear"*), and then finetune Gaussians based on the edited 2D images. We compare our method with Instruct-GS2GS (Vachha & Haque, 2024), which is also a 3DGS-based style transfer method.

## B ADDITIONAL EXPERIMENTAL RESULTS

### B.1 SCENE EDITING

**3D object inpainting.** Based on the 3D object removal, we could further accomplish the task of inpainting. Inspired by Gaussian Grouping (Ye et al., 2023), we first detect the noisy area after object removal through the mask association by DEVA (Cheng et al., 2023), and then use LAMA (Suvorov et al., 2022) to inpaint the noisy deletion area for each view. Finally, we finetune Gaussians with similar training process according to the 2D inpainted images for about 5 minutes to get the 3D inpainting results. Fig. 8 demonstrates the effectiveness of our method for 3D object inpainting.

**3D object style transfer.** By selecting an object (similar to object removal), we could also achieve 3D object style transfer. First, we select all correlated Gaussians of the object through 3D segmentation. Second, we freeze the parameters of other Gaussians and start finetuning. During finetuning, we use the edited 2D ground truth images as supervision which are produced by Instruct-Pix2Pix (Brooks et al., 2023), updating the original dataset. The pipeline is the same as Instruct-GS2GS. During finetuning, We employ L1 loss inside the 2D mask of the selected object and LPIPS loss within the bounding box that encloses the mask. From Fig. 9, we could see that our method significantly reduced the influence of the area that is not related to the targeted object. Instruct-GS2GS changes the hue of the whole figure and produce artifacts after editing.

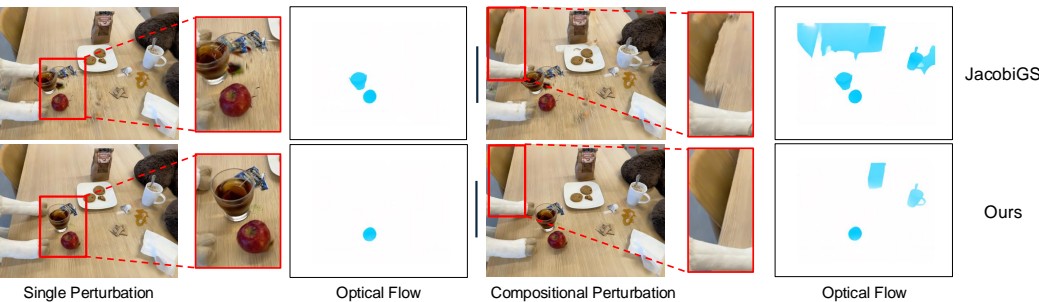

Figure 10: Visual comparison of multi-object editing between JacobiGS (Xu et al., 2023) and ours, where we slightly perturb three different objects: apple, cup, and bag. We use RAFT (Teed & Deng, 2020) to estimate the optical flow for visualizing the movement.

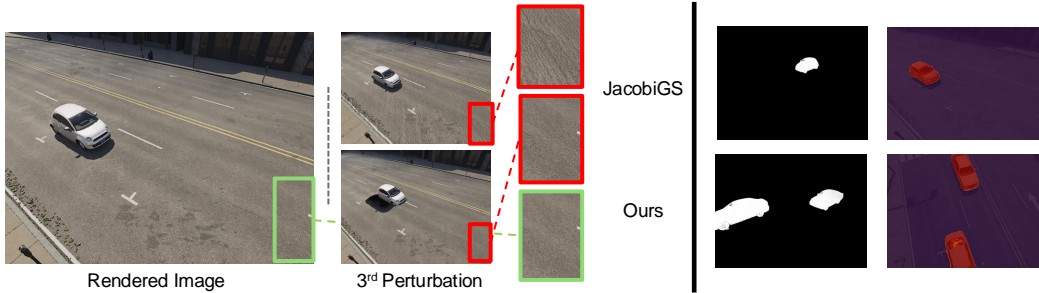

Figure 11: Qualitative results on NeRDS 360 (Irshad et al., 2023). Left: Comparison of consecutive editions when moving the car. Right: The gallery of segmentation mask and relevance map from different views(when perturbing a single Gaussian), obtained by our method.

**Multi-object editing.** To achieve edit multiple objects in the same scene, we simply add Jacobians of selected Gaussians of different objects to the network parameters. As shown in Fig. 10, JacobiGS has difficulty maintaining scene integrity when combining Jacobians of two or more objects for perturbations. For instance, the table nearby is also affected by the perturbation (see the red box comparison of the right part in Fig. 10). Conversely, our method effectively maintains robust correlation shaping, demonstrating superior performance in preserving the scene under multi-object editing.

## B.2 OUTDOOR SCENARIOS

To test the generalizability of our method, we employ our shaping method in NeRDS 360 (Irshad et al., 2023), which is an outdoor unbounded dataset comprising 75 unbounded and diverse scenes. From Fig. 11(a), we can clearly see that our method moves the car without creating any artifacts. JacobiGS fails to make the car move while changing the texture of the surrounding road. Fig. 11(b) illustrated the ability to detect objects like cars in the 3D space, showing the potential of our method to be applied in practical scenarios.

## B.3 ABLATION STUDIES

**Robustness of correlation shaping.** Though the masks generated by SAM (Kirillov et al., 2023) with DEVA (Cheng et al., 2023) are associated from different views, this strategy still suffers from the multi-granularity across different views and 2D segmentation errors. Fig. 12 demonstrates that our method resolves these problems by shaping the correlations between 3D Gaussians. Given a small part of the object as the prompt, our method can produce the segmentation of the entire object, showing that Gaussians inside the object are greatly associated.

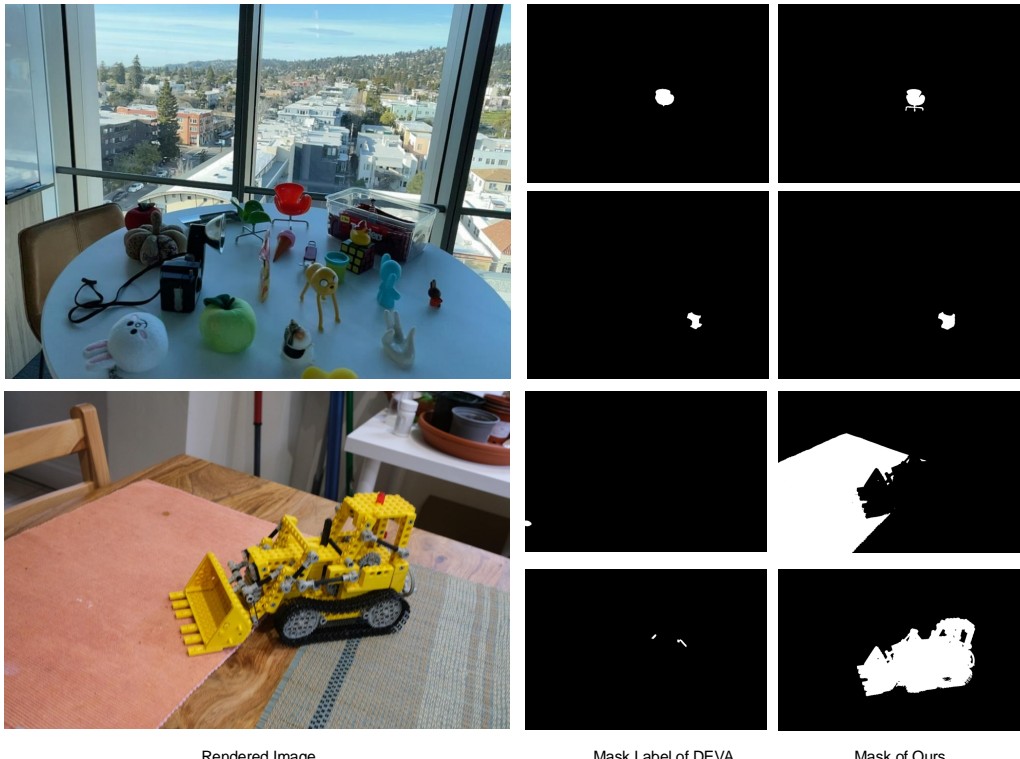

Rendered Image                     Mask Label of DEVA          Mask of Ours

Figure 12: Robustness of our method to the multi-granularity and errors of masks. When masks from supervision contain errors (e.g., missing legs of the chair) with multi-granularity (e.g., a small part of the tractor is segmented in a certain view), our correlation shaping successfully addresses these problems and produces correct segmentation masks.

**Quality of our 3D labels.** We visualize the quality of 3D labels in Fig.13, which are generated using the coarse mask labeling method from Eq.9. As illustrated, these 3D labels are consistent across multiple views (Fig.13(a)) and exhibit high quality (Fig.13(b)) without requiring any optimization.

**Regularization loss.** During finetuning the attribute decoding network, we introduce an extra regularization loss to help shape the Jacobians. Here Fig. 14 shows the ablation study of this regularization loss. From the area, the area in the red box apparently illustrates the impact of the regularization loss. When perturbing a Gaussian of the tractor, the one with regularization loss can maintains the minimal affects to other objects, while the one without it produces some artifacts to the surroundings. It demonstrates that the regularization loss can better shape the correlation structure of the tangent space of the motion network.

## C   PROOF OF DERIVATION

Given that the attribute decoding networks in neural fields (e.g., NeRFs and 3DGS) are MLPs, we focus on perturbing the weights $W^{(l)}$ of the $l$-th linear layer in the network $\mathbf{\Phi}_a$ (i.e., $\theta_D = W^{(l)}$), expressed as: $h^{(l)} = W^{(l)}\sigma(h^{(l-1)}) + b^{(l)}$. Let $\sigma$ denote the non-linear activation function, $b^{(l)}$ be the bias, and $h^{(l)}$ be the (hidden) output of the $l$-th layer.

To expose the relationship between the gradient of a Gaussian $\mathbf{g}_i$, $\partial\mathbf{\Phi}_i = \frac{\partial\mathbf{\Phi}(f(\mathbf{g}_i);\theta)}{\partial\theta_D}$, and the perturbed weight $W^{(l)}$, we consider the gradient of an arbitrary scalar output $z$ in the attribute decoding network with respect to $W^{(l)}$:

$$z = Y(h^{(l)}) = Y(W^{(l)}\sigma(h^{(l-1)}) + b^{(l)}), \tag{13}$$

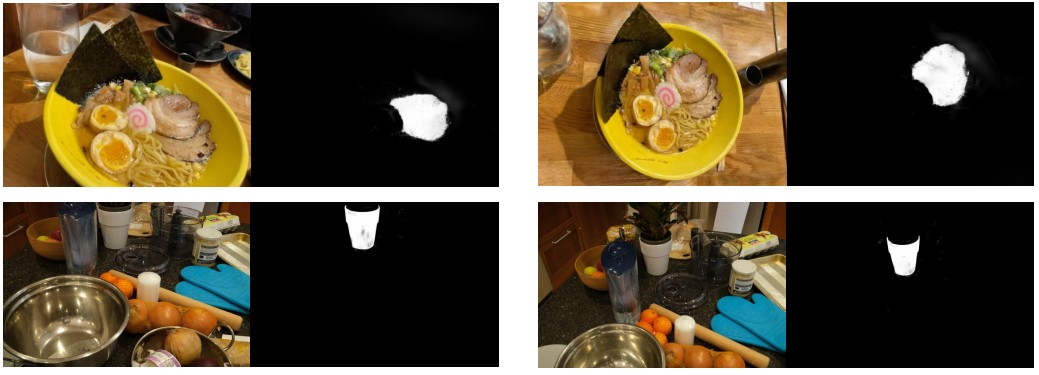

(a) Rendered images and corresponding 3D labels generated by coarse mask labeling

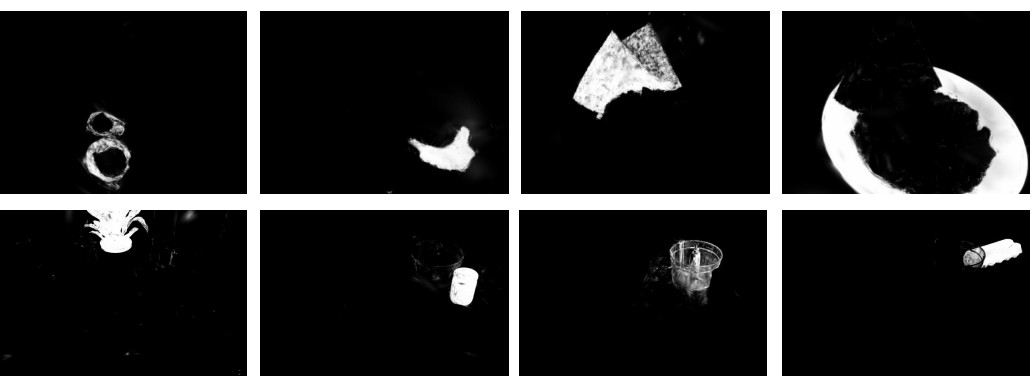

(b) More labels generated by coarse mask labeling

Figure 13: (a) Without any optimization, we produce 3D labels that are consistent across different views. (b) More examples of our 3D labels.done

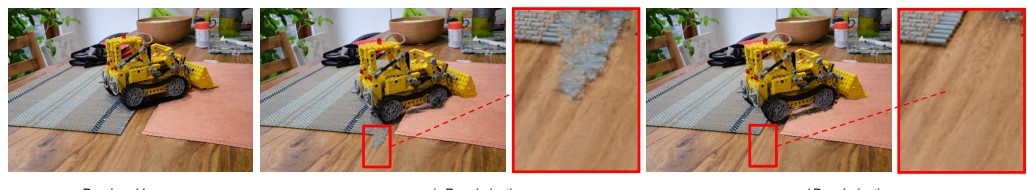

Rendered Image    w/o Regularization    w/ Regularization

Figure 14: The impact of regularization loss when moving the lego tractor.

where $Y$ denotes all the transformations after $l$-th layer, $z$ denotes an arbitrary scalar in the vector output of $\mathbf{\Phi}$. Then the differential $\mathrm{d}z$ could be written as:

$$\mathrm{d}z = \frac{\partial z}{\partial h^{(l)}}^{\top} \mathrm{d}h^{(l)} = \sigma(h^{(l-1)})\frac{\partial z}{\partial h^{(l)}}^{\top} \mathrm{d}W^{(l)} = \mathrm{tr}(\sigma(h^{(l-1)})\frac{\partial z}{\partial h^{(l)}}^{\top} \mathrm{d}W^{(l)}), \qquad (14)$$

as $\mathrm{d}z = \mathrm{tr}(\frac{\partial z}{\partial W^{(l)}}^{\top} \mathrm{d}W^{(l)})$, we could get that:

$$\frac{\partial z}{\partial W^{(l)}} = \frac{\partial z}{\partial h^{(l)}} \sigma(h^{(l-1)})^{\top}, \qquad (15)$$

which is a 2D matrix that has the same shape with the $W^{(l)}$. Then the cosine similarity of Jacobians $\partial\mathbf{\Phi}$ between two Gaussians are:

$$\cos(\partial\mathbf{\Phi}_i, \partial\mathbf{\Phi}_j) = \frac{\left(\frac{\partial z(\mathbf{x}_i)}{\partial h^{(l)}(\mathbf{x}_i)}\sigma(h^{(l-1)}(\mathbf{x}_i))^{\top}\right) \cdot \left(\frac{\partial z(\mathbf{x}_j)}{\partial h^{(l)}(\mathbf{x}_j)}\sigma(h^{(l-1)}(\mathbf{x}_j))^{\top}\right)}{\|\partial\mathbf{\Phi}_i\|\|\partial\mathbf{\Phi}_j\|}$$
$$= \frac{\left(\frac{\partial z(\mathbf{x}_i)}{\partial h^{(l)}(\mathbf{x}_i)} \cdot \frac{\partial z(\mathbf{x}_j)}{\partial h^{(l)}(\mathbf{x}_j)}\right)\left(\sigma(h^{(l-1)}(\mathbf{x}_i))^{\top} \cdot \sigma(h^{(l-1)}(\mathbf{x}_j))^{\top}\right)}{\|\partial\mathbf{\Phi}_i\|\|\partial\mathbf{\Phi}_j\|}, \qquad (16)$$

because $\partial h = \frac{\partial h^{(l)}}{\partial W^{(l)}}$ could be formulated as a 3D tensor, which could be conceptualized as a repeated collection of gradients and each gradient is the vector $\sigma(h^{(l-1)})$. If $\sigma(h^{(l-1)})$ is *well-shaped* satisfying object-level mutual relations (i.e., the absolute value of cosine similarity of $\sigma(h^{(l-1)})$ approaches 1 if $\mathbf{g}_i$ and $\mathbf{g}_j$ belong to the same object, or approaches 0 if not), then $\partial h$ should also satisfy. In this case, if $g_i$ and $g_j$ are affiliated to different objects:

$$
\begin{aligned}
\cos(\partial \mathbf{\Phi}_i, \partial \mathbf{\Phi}_j) &= \frac{(\frac{\partial z(\mathbf{x}_i)}{\partial h^{(l)}(\mathbf{x}_i)} \cdot \frac{\partial z(\mathbf{x}_j)}{\partial h^{(l)}(\mathbf{x}_j)})(\sigma(h^{(l-1)}(\mathbf{x}_i))^\top \cdot \sigma(h^{(l-1)}(\mathbf{x}_j))^\top)}{\|\partial \mathbf{\Phi}_i\|\|\partial \mathbf{\Phi}_j\|} \\
&= \frac{(\frac{\partial z(\mathbf{x}_i)}{\partial h^{(l)}(\mathbf{x}_i)} \cdot \frac{\partial z(\mathbf{x}_j)}{\partial h^{(l)}(\mathbf{x}_j)})(0)}{\|\partial \mathbf{\Phi}_i\|\|\partial \mathbf{\Phi}_j\|} \\
&= 0,
\end{aligned}
\tag{17}
$$

Eq. 17 indicates that if $\cos(\partial h_i, \partial h_j) = 0$ when $g_i$ and $g_j$ are affiliated to different objects, then $\cos(\partial \mathbf{\Phi}_i, \partial \mathbf{\Phi}_j) = 0$. Meanwhile, $\sigma(h^{(l-1)})$ would not change since the input of $\mathbf{\Phi}_a$ is the fixed $(f(\mathbf{g}_i), d)$. Thus $\cos(\partial h_i, \partial h_j) = 0$ would be consistent during the perturbation. Therefore, we could have the following lemma:

**Lemma 1.** (*Orthogonality*) After any number of perturbations, the orthogonality of Jacobians between two Gaussians $g_i \in \mathcal{G}_A$ and $g_j \in \mathcal{G}_B$ that are affiliated to different objects ($\mathcal{G}_A \neq \mathcal{G}_B$) would remain consistent:

$$
\cos(\partial \mathbf{\Phi}_i^{(d)}, \partial \mathbf{\Phi}_j^{(d)}) = \cos(\partial h_i^{(d)}, \partial h_j^{(d)}) = \cos(\partial h_i^{(0)}, \partial h_j^{(0)}), d \in \mathbb{N}^+,
\tag{18}
$$

In contrast, we have another lemma:

**Lemma 2.** (*Similarity*) After any number of perturbations, the similarity of Jacobians between two Gaussians $g_i \in \mathcal{G}_A$ and $g_j \in \mathcal{G}_B$ that are affiliated to the same objects ($\mathcal{G}_A = \mathcal{G}_B$) would remain consistent:

$$
\cos(\partial \mathbf{\Phi}_i^{(d)}, \partial \mathbf{\Phi}_j^{(d)}) \approx \cos(\partial h_i^{(d)}, \partial h_j^{(d)}) = \cos(\partial h_i^{(0)}, \partial h_j^{(0)}), d \in \mathbb{N}^+,
\tag{19}
$$

*Proof of Lemma 2*: Consider the set of Jacobians $J_A = \{\partial \mathbf{\Phi}_i, g_i \in \mathcal{G}_A\}$ of all Gaussians in an arbitrary object $\mathcal{G}_A$, they all would be orthogonal to the Jacobians of Gaussians from other objects, according to **Lemma 1** if $\partial h$ is *well-shaped*. Thus, the representation space of $J_A$ would be very narrow. If the representation space of $\partial \mathbf{\Phi}$ is compact enough (e.g., the dimension of the representation space of $\partial \mathbf{\Phi}$ is identical to the number of objects in the scene), then the similarity $\cos(\partial \mathbf{\Phi}_i^{(d)}, \partial \mathbf{\Phi}_j^{(d)}) = 1$ when $\mathcal{G}_A = \mathcal{G}_B$. In this extreme case, the set of Jacobians from all the objects in the scene is a group of canonical basis (Bronson, 1969) of the representation space of $\partial \mathbf{\Phi}$.

Combining **Lemma 1** and **Lemma 2**, we could have the conclusion in Eq. 6.

