# OpenReview forum: "InfoGS: Efficient Structure-Aware 3D Gaussians via Lightweight Information Shaping"
_ICLR.cc/2025/Conference — ICLR 2025 Poster_

### Official Review · Reviewer_fawD · 2024-10-30

**Soundness:** 3
**Presentation:** 3
**Contribution:** 3
**Rating:** 6
**Confidence:** 3

**Summary:**

This paper introduces a mutual information technique based on 3D Gaussian splatting. Building upon JacobiNeRF, the authors present a more efficient training pipeline that optimizes activations instead of Jacobians. This approach ensures that the correlations between Gaussians are correctly shaped and remain consistent through successive parameter changes. Experiments on open-vocabulary segmentation and scene editing demonstrate the efficiency and significant capabilities of the proposed method.

**Strengths:**

1. The paper is well-written and organized, the authors provide a thorough theoretical analysis of the proposed method.
2. Experiments demonstrate that the proposed method outperforms other baselines while maintaining robustness to incorrect masks.

**Weaknesses:**

1. Compared to methods that edit the scene after obtaining the labels for each Gaussian, such as Gaussian Grouping, the direct manipulation of network parameters appears less flexible. For instance, while Gaussian Grouping can achieve style transfer, the proposed method seems limited to modifying the colors of selected Gaussians to be uniform.

2. The authors conduct open-vocabulary segmentation in only three scenes, evaluating just one object per scene if I understand correctly. Including more experiments would enhance the robustness and credibility of the results.

**Questions:**

1. Aside from JacobiGS in the paper, which utilizes the same mutual information shaping loss as JacobiNeRF, why was the original JacobiNeRF not included as a baseline for comparison?


2. Regarding weakness 2, is it possible for the proposed method to edit the color of an object to a textured color? Additionally, for the object movement task, can the method modify the trajectory of a selected object to follow a position-related trajectory, such as rotation?


3. Besides open-vocabulary segmentation tasks, would it be feasible to conduct experiments on semantic or instance segmentation tasks to further enhance the credibility of the findings?

---

> ### Author Response · Authors · 2024-11-20
> **Response to weaknesses and questions (Part 1)**
>
> We sincerely appreciate the reviewer for the valuable feedback on our work, and we provide our responses to the concerns below. We hope our response addresses the issues raised in the review.
>
> * **W.1 & Q.2**: Compared to methods that edit the scene after obtaining the labels for each Gaussian, such as Gaussian Grouping, the direct manipulation of network parameters appears less flexible. For instance, while Gaussian Grouping can achieve style transfer, the proposed method seems limited to modifying the colors of selected Gaussians to be uniform. Regarding weakness 2, is it possible for the proposed method to edit the color of an object to a textured color? Additionally, for the object movement task, can the method modify the trajectory of a selected object to follow a position-related trajectory, such as rotation?
>
> **A**: Our method can support the task of style transfer and texture editing. We provide qualitative results and a detailed process in Appendix B (see Line 857) and outperform Instruct-GS2GS [1], which is a 3D style transfer editing method based on 3DGS. Figure 9 clearly demonstrates that our method significantly reduces the influence on areas not related to the targeted object, compared to Instruct-GS2GS. The flexibility of our method is no less than that of Gaussian Grouping, as Gaussian Grouping relies on object selection for various editing tasks. In 3D object style transfer, Gaussian Grouping selects the Gaussians of the object and edits 2D images produced from [3] to finetune. We can also select the Gaussians of the object through our 3D segmentation and finetune on edited 2D images, similar to changing the texture of the object. We are happy to provide more examples to illustrate the flexibility of our method.
>
> Regarding rotation, a rigid rotation can be achieved through a manual calculation of the object [5], using SE(3) poses. However, it is challenging to imitate flexible position-related trajectories, such as deformation. As discussed in our limitations, we plan to explore this in our future work by learning position-related correlations from videos.
>
> *  **W.2 & Q.3**: The authors conduct open-vocabulary segmentation in only three scenes, evaluating just one object per scene if I understand correctly. Including more experiments would enhance the robustness and credibility of the results. Besides open-vocabulary segmentation tasks, would it be feasible to conduct experiments on semantic or instance segmentation tasks to further enhance the credibility of the findings?
>
> **A**: There are an average of eight objects per scene in the LERF-Localization dataset [4], which is sufficient to demonstrate segmentation performance. Besides open-vocabulary segmentation, we additionally provide semantic segmentation experimental results on [2], which provides manually labeled segmentation masks on MipNeRF-360 and LERF dataset. The table below shows that our method outperforms Gaussian Grouping and other baselines with the least training cost, further indicating the credibility of our segmentation capability.
> | Model | MipNeRF-360 | LERF |
> |---------|---------|---------|
> | JacobiNeRF   |  85.2  | 62.7   |
> | SA3D   | 88.8   | 69.3   |
> |  JacobiGS  | 91.7   |   90.9|
> | SANeRF-HQ | 91.0  | 90.7   |
> | Gaussian Grouping | 93.1   | 91.5   |
> | Ours   | 93.7   | 92.9  |
>
> **Reference**:
>
> [1] Cyrus Vachha and Ayaan Haque. Instruct-gs2gs: Editing 3d gaussian splats with instructions, 2024.
>
> [2] Yichen Liu, Benran Hu, Chi-Keung Tang, and Yu-Wing Tai. Sanerf-hq: Segment anything for nerf in high quality.
>
> [3] Tim Brooks, Aleksander Holynski, and Alexei A Efros. Instructpix2pix: Learning to follow image editing instructions. In Proceedings of the IEEE/CVF Conference on Computer Vision and PatternRecognition, pp. 18392–18402, 2023.
>
> [4] Mingqiao Ye, Martin Danelljan, Fisher Yu, and Lei Ke. Gaussian grouping: Segment and edit anything in 3d scenes. arXiv preprint arXiv:2312.00732, 2023.
>
> [5] Kerr, Justin, et al. "Robot see robot do: Imitating articulated object manipulation with monocular 4d reconstruction." arXiv preprint arXiv:2409.18121 (2024).
>
> [6] Jiazhong Cen, Zanwei Zhou, Jiemin Fang, Wei Shen, Lingxi Xie, Dongsheng Jiang, Xiaopeng Zhang, Qi Tian, et al. Segment anything in 3d with nerfs. Advances in Neural Information Processing Systems, 36:25971–25990, 2023.

---

> ### Author Response · Authors · 2024-11-20
> **Response to weaknesses and questions (Part 2)**
>
> * **Q.1**: Aside from JacobiGS in the paper, which utilizes the same mutual information shaping loss as JacobiNeRF, why was the original JacobiNeRF not included as a baseline for comparison?
>
> **A**: We have added the quantitative results of JacobiNeRF in Table 1 (see the revised paper). The performance of JacobiNeRF is close to SA3D[6], a similar NeRF-based segmentation work. There is a significant gap between it and our method due to poor reconstruction quality. We did not include JacobiNeRF as a baseline in our original submission because JacobiNeRF builds on plain NeRF. The reconstruction ability of JacobiNeRF is insufficient for complex open-world unbounded scenes, which are used in our experiments. Subsequent works like MipNeRF were proposed to address these issues. Moreover, 3DGS is a more advanced fundamental framework for reconstruction. For fair comparisons, we used JacobiGS to mitigate the reconstruction gap and solely compare the performance of correlation shaping on 3D editing. JacobiGS utilizes the identical training pipeline and framework as ours, with the exception that the correlation shaping loss is sourced from JacobiNeRF.
>
>
> In summary, we demonstrated the flexibility of our method in terms of 3D editing by providing results of 3D object style transfer. We also explained the concerns related to the segmentation dataset we used, and additionally compared the performance on another semantic segmentation task. We also included JacobiNeRF as a baseline in the segmentation task and clarified that it is not considered in our original submission due to the unfair comparison.
>
> We hope our response addresses the concerns and assists in re-evaluating our work. Please feel free to let us know if you have further questions or comments.
>
> **Reference**:
>
> [1] Cyrus Vachha and Ayaan Haque. Instruct-gs2gs: Editing 3d gaussian splats with instructions, 2024.
>
> [2] Yichen Liu, Benran Hu, Chi-Keung Tang, and Yu-Wing Tai. Sanerf-hq: Segment anything for nerf in high quality.
>
> [3] Tim Brooks, Aleksander Holynski, and Alexei A Efros. Instructpix2pix: Learning to follow image editing instructions. In Proceedings of the IEEE/CVF Conference on Computer Vision and PatternRecognition, pp. 18392–18402, 2023.
>
> [4] Mingqiao Ye, Martin Danelljan, Fisher Yu, and Lei Ke. Gaussian grouping: Segment and edit anything in 3d scenes. arXiv preprint arXiv:2312.00732, 2023.
>
> [5] Kerr, Justin, et al. "Robot see robot do: Imitating articulated object manipulation with monocular 4d reconstruction." arXiv preprint arXiv:2409.18121 (2024).
>
> [6] Jiazhong Cen, Zanwei Zhou, Jiemin Fang, Wei Shen, Lingxi Xie, Dongsheng Jiang, Xiaopeng Zhang, Qi Tian, et al. Segment anything in 3d with nerfs. Advances in Neural Information Processing Systems, 36:25971–25990, 2023.

---

> > ### Comment · Reviewer_fawD · 2024-11-25
> >
> > I thank the authors for their time and effort in addressing my concerns. All of my questions have been resolved.

---

> > > ### Author Response · Authors · 2024-11-26
> > > **Gratitude for your feedback**
> > >
> > > Thank you for acknowledging that the additional experiments and clarifications addressed your concerns. We sincerely appreciate your thorough review and thoughtful feedback, which have significantly contributed to the improvement of our work. We hope the additional results convincingly demonstrate the robustness and impact of our approach. Should you require any further information to finalize your evaluation, please do not hesitate to let us know.

---

### Official Review · Reviewer_fQfp · 2024-11-02

**Soundness:** 3
**Presentation:** 1
**Contribution:** 2
**Rating:** 6
**Confidence:** 4

**Summary:**

This paper proposes information shaping for the 3D Gaussian field, following NeRF-based work JacobiNeRF. By contrastive learning between Gaussian primitives, the resulting Gaussian primitives on the same entities have a strong correlation while primitives on different entities have a weak correlation. The technical part follows the basic idea of JacobiNeRF and is tailored to 3D Gaussian with a attributes decoding network.  The learned Gaussian field is easier to edit, such as object removal and object movement.

**Strengths:**

- This paper adapts the basic idea of JacobiNeRF to 3DGS and makes it work in terms of scene editing.
- The method achieves good performance in the open vocabulary segmentation task.
- The application of object removal seems have good qualitative performance.

**Weaknesses:**

- The technical novelty is limited. The Jacobian-based mutual information learning is the core part of this paper, which mainly follows JacobiNeRF. The authors are suggested to emphasize the unique designs that are different from the JacobiNeRF and demonstrate the effectiveness of the unique designs. The mask guidance and smoothness regularization differ from the DINO adopted in JacobiNeRF. However, these components seem to be incremental techniques, and their effectiveness is not well validated in the experiments.
- The writing should be improved. Many concepts lack a clear definition. For example, what are the exact definitions of perturbation and the sequential of perturbations (e.g., 1st/2nd/3rd perturbation). What are the exact parameters perturbated and how are they perturbated (i.e., perturbation direction and norm).
- The demonstration is weak. The object movement in Figure 6 is not impressive and it seems to be a sample handpicked with effort. It would be better to present more object movement demonstrations, using the main components of a 3D scene, such as the earthmover toy in the kitchen scene. Can you move the earthmover to anywhere else on the desk? Furthermore, the demonstrations have low diversity, mainly adopting the scene of kitchen and bear.
- The method outperforms JacobiNeRF and JacobiGS. However, the authors do not show the reasons via an apple-to-apple comparison with JacobiNeRF and JacobiGS. As I suggested in the first item, the authors are encouraged to show the performance roadmap from the JacobiGS baseline (or other reasonable baseline) and add the unique components one by one, offering a better understanding of the inner workings to readers.

**Questions:**

Does the decoding network reduce the rendering efficiency?

---

> ### Author Response · Authors · 2024-11-20
> **Response to weaknesses and questions (Part 1)**
>
> We thank the reviewer for their valuable time and insightful comments. We hope our responses can address your concerns and better clarify the contributions and value of our work.
>
> * **Weakness.1,4**: The technical novelty is limited. The Jacobian-based mutual information learning is the core part of this paper, which mainly follows JacobiNeRF. The authors are suggested to emphasize the unique designs that are different from the JacobiNeRF and demonstrate the effectiveness of the unique designs. The mask guidance and smoothness regularization differ from the DINO adopted in JacobiNeRF. However, these components seem to be incremental techniques, and their effectiveness is not well validated in the experiments.
> The method outperforms JacobiNeRF and JacobiGS. However, the authors do not show the reasons via an apple-to-apple comparison with JacobiNeRF and JacobiGS. As I suggested in the first item, the authors are encouraged to show the performance roadmap from the JacobiGS baseline (or other reasonable baseline) and add the unique components one by one, offering a better understanding of the inner workings to readers.
>
> **Answer.1,4**: The core contribution is indeed correlation shaping for editing, but the proposed shaping framework is largely derived from JacobiNeRF.
> 1. First, we proposed correlation shaping for 3D editing based on 3DGS, whereas JacobiNeRF proposed to do label propagation for 2D pixels based on NeRF. NeRF and 3DGS are two different scene representations, and the theoretical derivations of mutual information under these two frameworks, for different goals, are also different.
> 2. Second, transferring the shaping method from JacobiNeRF to 3DGS does not work. We introduced an extra component of the attribute decoding network, enabling us to associate independent Gaussians by optimizing the network through correlation shaping. **By applying a similar correlation shaping loss from JacobiNeRF to this new 3DGS framework, along with supervision refinement (video tracker and smoothness loss), we developed the baseline named JacobiGS**. However, as shown in the object movement task (Figure 6 and more examples in the supplementary materials), JacobiGS cannot satisfy the need for consecutive editing operations. This is because the previous approach in JacobiNeRF uses a point-wise correlation shaping method (see Line 249). In light of this, we theoretically prove (see Appendix C) that the proposed new shaping loss ensures consistency among Gaussians after consecutive parameter changes, thus supporting versatile editing. **By solely replacing the shaping loss in JacobiGS with ours**, we demonstrate its effectiveness through extensive experiments.
> 3. Furthermore, the shaping loss in JacobiNeRF is constrained to optimize the layers near the network output; otherwise, the time and memory cost of calculating Jacobians and higher-order gradients during backpropagation becomes unaffordable. **In practice, optimizing the Jacobian of the second layer and sampling only 128 Gaussians for contrastive learning exceeds the storage capacity of a single A100 GPU**. A detailed roadmap is also discussed in the general response, common issue 1.
>
> In conclusion, the uniqueness of our method lies in providing a theoretical analysis for consistent mutual information shaping, along with proposing a training pipeline with lightweight optimization on a novel 3DGS framework, yielding superior performance in 3D editing tasks.
>
>
>
> * **Weakness.2**: The writing should be improved. Many concepts lack a clear definition. For example, what are the exact definitions of perturbation and the sequential of perturbations (e.g., 1st/2nd/3rd perturbation). What are the exact parameters perturbated and how are they perturbated (i.e., perturbation direction and norm).
>
> **Answer.2**: Thank you for pointing out this concern. Perturbation here refers to parameter changes according to the selected Gaussian. Specifically, a perturbation involves adding the multiplication of the Jacobian $\partial\Phi_i$ of selected Gaussian $g_i$ with respect to the optimized layer and $\sigma_s$. $\sigma_s$ is a scaling factor that controls the impact of the change. We have clarified this in the revised paper (please refer to Line 482). We hope that the revised part alleviates the confusion, and we are happy to provide more details if needed.

---

> ### Author Response · Authors · 2024-11-20
> **Response to weaknesses and questions (Part 2)**
>
> * **Weakness.3**: The demonstration is weak. The object movement in Figure 6 is not impressive and it seems to be a sample handpicked with effort. It would be better to present more object movement demonstrations, using the main components of a 3D scene, such as the earthmover toy in the kitchen scene. Can you move the earthmover to anywhere else on the desk? Furthermore, the demonstrations have low diversity, mainly adopting the scene of kitchen and bear.
>
> **Answer.3**: We provide more examples of consecutive editing from different views and scenes in the supplementary materials for the comparison between JacobiGS and our method, due to page limit. The movement of the earthmover is also included, and the movement can be arbitrary. Since parameter perturbation does not affect the network output of uncorrelated Gaussians, we apply a 3x3 diagonal matrix to redirect the change of the network outputs, allowing the object to be moved anywhere. An illustration of this case is provided in the figurine scene in the supplementary materials, where the direction of the pumpkin changes twice during movement. We are happy to upload more examples to demonstrate the diversity and validity of our method if not enough.
>
>
>
> * **Question.1**: Does the decoding network reduce the rendering efficiency?
>
> **Answer**: The decoding network does not affect the rendering efficiency. The attributes of 3D Gaussians are all precomputed before rendering, which means that the network only needs to perform inference once. This single pass is very fast since the network is small. Therefore, our method has the same FPS as the original 3DGS. The render speed of our method is 120 FPS on the MipNeRF-360 dataset, the same as the original 3DGS.
>
> In summary, we have addressed the weaknesses and questions raised in your review. We have explained the major theoretical and technical innovations introduced in the previous 3DGS framework, focusing on exploiting correlations to associate independent Gaussians for interactive tasks. We also clarified misunderstandings related to the writing and experiments. We carefully revised the related parts and provided more examples to enhance the credibility of our method.
>
> We hope our response addresses the concerns and assists in re-evaluating our work. Please feel free to let us know if you have further questions or comments

---

> > ### Comment · Reviewer_fQfp · 2024-11-25
> > **Further questions**
> >
> > Thanks for your response. I have further questions.
> > 1. How do you handle the semantic ambiguity? Can you move an independent part of an object instead of moving the whole object? How do you control the editing granularity?
> > 2. The demonstrated object movement seems a little trivial. It may be directly implemented by instance segmentation from multiple views.

---

> > > ### Author Response · Authors · 2024-12-02
> > > **Appreciation for Your Revised Evaluation and Feedback**
> > >
> > > Dear Reviewer,
> > >
> > > Thank you for taking the time to carefully assess our rebuttal and for raising your score following our clarifications and additional experiments. We are grateful for your thoughtful review and feedback, which have been instrumental in improving our work.
> > >
> > > We are pleased that our response addressed the key concerns you raised, as evidenced by your updated evaluation. The additional experiments and clarifications provided were specifically designed to address these points, and we hope they demonstrate the robustness and significance of our approach.
> > >
> > > Should you have any remaining questions or require further details, we are more than happy to provide additional information. Thank you again for your valuable input and support of our work.
> > >
> > > Sincerely,
> > >
> > > Authors

---

> ### Author Response · Authors · 2024-11-26
> **Response to further questions**
>
> * **Question 1**: How do you handle the semantic ambiguity? Can you move an independent part of an object instead of moving the whole object? How do you control the editing granularity?
>
> **Answer**: 1. Our method is compatible with other segmentation techniques that produce finer-grained segmentations. By leveraging these more detailed segmentations, we can shape the network to manipulate smaller object parts. We follow previous works[1, 2, 3] on generating supervision masks to control the editing granularity. In our paper, we show that our method works with ground truth segmentation masks (Appendix B.2), those provided by SAM [4] (experiments in the main paper). We can move smaller objects by adjusting the hyper-parameters setting when using SAM. We can also use SAM features as supervision, and raise the threshold of feature similarity during contrastive learning to get more fine-grained results.
>
> To address semantic ambiguity, more accurate segmentation masks without semantic overlaps can be employed.  To move an independent part of an object instead of moving the whole object, we need to first shape our network with segmentation masks that can disambiguate parts, and then we can move each individual part. To control the editing granularity, it is now determined by the segmentation granularity. However, since our method is compatible with different segmentation masks, we can perform more granular edits by shaping the scene with more granular segmentation supervision.
>
> We are happy to provide a more detailed discussion in the paper to further emphasize these points.
>
> * **Question 2**: The demonstrated object movement seems a little trivial. It may be directly implemented by instance segmentation from multiple views.
>
> Moving objects in a scene represented by 3DGS is non-trivial for several reasons.
>
> First, while object movement can be achieved through instance segmentation (e.g., Gaussian grouping) from multiple views, our approach outperforms this baseline. To quantify this, we compare the rendered images after moving the object with the original image using the LPIPS [5] metric on MipNeRF-360 and LERF datasets. As shown in the table below, our method consistently results in higher image quality after editing.
>
> | Model | MipNeRF-360 | LERF |
> |---------|---------|---------|
> | Gaussian Grouping | 0.225  | 0.217   |
> | Ours   | 0.183   | 0.164  |
>
> Second, segmentation-based methods for object movement are less efficient. These methods require computing feature similarities between 3D Gaussians and given prompts (e.g., text label, 2D-pixel prompt, or other possible labels), segmenting the object, and then performing the movement. This is inefficient for scenes containing millions of Gaussians—a typical scenario for large, unbounded scenes, especially in editing multiple objects. In contrast, our correlation-based approach simplifies scene editing by adjusting network parameters once to directly perform versatile editing. Our method eliminates the need for additional computations and significantly reduces both time and memory requirements.
>
> We hope the clarifications and additional experiments addressed your concerns. Please let us know if there are any remaining aspects you feel require further clarification or elaboration.
>
> **Reference**:
>
> [1] Zhou, Shijie, et al. "Feature 3dgs: Supercharging 3d gaussian splatting to enable distilled feature fields." Proceedings of the IEEE/CVF Conference on Computer Vision and Pattern Recognition. 2024.
>
> [2] Ye, Mingqiao, et al. "Gaussian grouping: Segment and edit anything in 3d scenes." European Conference on Computer Vision. Springer, Cham, 2025.
>
> [3] Hu, Xu, et al. "Semantic anything in 3d gaussians." arXiv preprint arXiv:2401.17857 (2024).
>
> [4] Kirillov, Alexander, et al. "Segment anything." Proceedings of the IEEE/CVF International Conference on Computer Vision. 2023.
>
> [5] Zhang, Richard, et al. "The unreasonable effectiveness of deep features as a perceptual metric." Proceedings of the IEEE conference on computer vision and pattern recognition. 2018.

---

### Official Review · Reviewer_WXPj · 2024-11-05

**Soundness:** 3
**Presentation:** 2
**Contribution:** 3
**Rating:** 6
**Confidence:** 4

**Summary:**

This paper presents a method for correlating Gaussians that belong to the same semantic group. It leverages a network to decode feature Gaussians into attributes and approximates mutual information via gradients with respect to network parameters. This approach enables editing operations through adjustments to the network parameters rather than direct manipulation of numerous Gaussians. The mutual information between Gaussians is approximated by shaping activations from perturbed network weights.

**Strengths:**

Previous approaches embed semantic features directly within each Gaussian primitive, requiring extensive manipulation of individual Gaussians to edit scenes. This paper instead proposes encoding mutual information within an attribute decoding network, allowing scene edits to be made by directly adjusting the network parameters.

**Weaknesses:**

The paper is generally challenging to follow, with many points lacking clear explanations. For example, Equation 7 and Line 304 state that "we find that ∂h corresponds to repeated activations σ(h(l−1))." However, this lacks sufficient context, as does Line 317, which mentions that "shaping activations ∂h within the attribute decoding network Φa supports a sequence of editing operations." Further explanation on why shaping these activations enables multiple edits would improve clarity.

The proposed method uses an MLP network to model mutual information between Gaussians, where the Gaussians primarily serve to splat features onto images used as inputs for the MLP. The primary contribution appears to be an enhancement to the JacobiNeRF framework, rather than a significant innovation in the context of 3D Gaussian Splatting (3DGS).

The contributions of Section 3.5 on coarse mask guidance and Section 3.6 on smoothness regularization seem minor, primarily focused on refining supervision masks. These sections could potentially be merged, as similar techniques have been explored in prior work.

The method depends on a 2D video mask tracker for grouping Gaussians. Based on the experimental results, it is difficult to determine if performance improvements arise from the mask tracker or the proposed approach. For instance, in Figure 4, the poor mask quality generated by Gaussian Grouping may stem from suboptimal 2D masks.

**Questions:**

Please refer to the weaknesses section.

---

> ### Author Response · Authors · 2024-11-20
> **Response to weaknesses and questions (Part 1)**
>
> We thank the reviewer for the positive feedback and acknowledgment of the uniqueness of our work compared to previous approaches. We hope our response addresses the issues raised in the review.
>
> * **Weakness.1,3**: The paper is generally challenging to follow, with many points lacking clear explanations. For example, Equation 7 and Line 304 state that "we find that ∂h corresponds to repeated activations σ(h(l−1))." However, this lacks sufficient context, as does Line 317, which mentions that "shaping activations ∂h within the attribute decoding network Φa supports a sequence of editing operations." Further explanation on why shaping these activations enables multiple edits would improve clarity.
> The contributions of Section 3.5 on coarse mask guidance and Section 3.6 on smoothness regularization seem minor, primarily focused on refining supervision masks. These sections could potentially be merged, as similar techniques have been explored in prior work.
>
>
> **Answer.1,3**: We apologize for the confusion and have edited the relevant parts in the revision (in blue). Regarding Equation 7 and Line 304, a detailed derivation and proof are provided in Appendix C, which we now reference in Line 304. We also appreciate the reviewer's suggestion regarding Sections 3.5 and 3.6. We have merged these sections to discuss mask supervision and related refinement. It is worth mentioning that lifting inconsistent 2D image masks into 3D Gaussian labels without optimization for refining supervision is rarely explored. Previous works primarily splat 3D Gaussians to planes to receive supervision from 2D masks. The ablation study on whether to use 3D labels is shown in the last two lines of Table 1, and the visualization of our 3D masks is provided in Figure 12 (Appendix B). Again, we are happy to revise other vague sections if additional suggestions are provided.
>
> * **Weakness.2**: The proposed method uses an MLP network to model mutual information between Gaussians, where the Gaussians primarily serve to splat features onto images used as inputs for the MLP. The primary contribution appears to be an enhancement to the JacobiNeRF framework, rather than a significant innovation in the context of 3D Gaussian Splatting (3DGS).
>
> **Answer.2**: We are committed to innovating the existing 3DGS framework. Previous 3DGS methods use a discrete representation, where each 3D Gaussian is independent of the others, resulting in unnecessary memory and time overhead [1, 2]. This issue becomes even more significant in 3D editing tasks, which require manipulating millions of Gaussians. To the best of our knowledge, we are the first to explore correlations and mutual information between discrete Gaussians to improve the efficiency of editing in the 3DGS field. The superior performance of our method, as demonstrated in Table 1, validates the effectiveness of our approach. This correlation-driven framework is entirely different from previous 3DGS methods, and many improvements can be built upon our framework, as discussed in the limitations. Regarding JacobiNeRF, NeRF is an implicit representation using MLP, whereas 3DGS is an explicit discrete representation. Although the conclusions of the mutual information derivation are similar, JacobiNeRF explores the changes of 2D pixels, directly optimizing the original NeRF MLP. Our method introduces an attribute decoding network upon basic 3DGS, and optimizes it to build consistent correlations among explicit 3D Gaussians. A detailed road map is also discussed in the general response, common issue 1.
>
> **Reference**
>
> [1] Tao Lu, Mulin Yu, Linning Xu, Yuanbo Xiangli, Limin Wang, Dahua Lin, and Bo Dai. Scaffold-gs: Structured 3d gaussians for view-adaptive rendering. ArXiv, abs/2312.00109, 2023
>
> [2] Yihang Chen, Qianyi Wu, Jianfei Cai, Mehrtash Harandi, and Weiyao Lin. Hac: Hash-grid assisted context for 3d gaussian splatting compression. ArXiv, abs/2403.14530, 2024.
>
> [3] Mingqiao Ye, Martin Danelljan, Fisher Yu, and Lei Ke. Gaussian grouping: Segment and edit anything in 3d scenes. arXiv preprint arXiv:2312.00732, 2023.

---

> ### Author Response · Authors · 2024-11-20
> **Response to weaknesses and questions (Part 2)**
>
> * **Weakness.4**: The method depends on a 2D video mask tracker for grouping Gaussians. Based on the experimental results, it is difficult to determine if performance improvements arise from the mask tracker or the proposed approach. For instance, in Figure 4, the poor mask quality generated by Gaussian Grouping may stem from suboptimal 2D masks.
>
> **Answer.4**: We have demonstrated in Table 1 that the major improvement comes from our shaping method. Gaussian Grouping uses the same 2D video mask tracker to group Gaussians [3], yet our method performs better in downstream tasks (see Table 1, Figures 4 and 5), using less memory and time for training. For a fair comparison, we used exactly the same 2D masks provided by Gaussian Grouping during training. One difference is that we use coarse mask labeling to obtain 3D labels based on the given 2D masks. However, as shown in the penultimate row of Table 1, our method still outperforms Gaussian Grouping without using the video mask tracker and 2D-to-3D lifting. This demonstrates that the improvements indeed arise from the brand new correlation shaping we proposed. We apologize for the confusion and have clarified this point in the revision (see Line 417).
>
> To recap, we have addressed the weaknesses raised in your review. We have explained the innovations introduced in our 3DGS framework, focusing on exploiting correlations to associate independent Gaussians for interactive tasks. We also clarified misunderstandings related to equations in our method and comparisons in the experiments.
>
> Please feel free to let us know if you have further questions or comments.
>
> **Reference**
>
> [1] Tao Lu, Mulin Yu, Linning Xu, Yuanbo Xiangli, Limin Wang, Dahua Lin, and Bo Dai. Scaffold-gs: Structured 3d gaussians for view-adaptive rendering. ArXiv, abs/2312.00109, 2023
>
> [2] Yihang Chen, Qianyi Wu, Jianfei Cai, Mehrtash Harandi, and Weiyao Lin. Hac: Hash-grid assisted context for 3d gaussian splatting compression. ArXiv, abs/2403.14530, 2024.
>
> [3] Mingqiao Ye, Martin Danelljan, Fisher Yu, and Lei Ke. Gaussian grouping: Segment and edit anything in 3d scenes. arXiv preprint arXiv:2312.00732, 2023.

---

> ### Author Response · Authors · 2024-12-02
>
> As the discussion deadline approaches, the authors kindly encourage the reviewer to review the responses. We have thoughtfully revised the manuscript based on the valuable suggestions and would greatly appreciate any further feedback!

---

### Author Response · Authors · 2024-11-20
**Response addressing all Reviewers (Part 1)**

We would like to thank the reviewers for the insightful feedback! We are glad that reviewers find our study “unique” compared to pervious approaches (Reviewer WXPj), “effective” (Reviewer fQfp, fawD), and “theoretically insightful” (Reviewer faWD). To recap, we propose a mutual information shaping technique that can coordinate correlated Gaussians via an attribute decoding net, and the scene could be edited by adjusting the network parameters. To the best of our knowledge, we are the first to explore the mutual information between Gaussians to boost the efficiency of editing in the 3DGS field. We demonstrate that our shaping technique can be more efficient than previous 3D editing approaches. Our method can outperform several baselines in tasks including removal, inpainting, and segmentation. **We hope our paper inspires innovative designs for 3D representations that effectively leverage correlation information, enabling interactive tasks requiring a deep understanding of the underlying scene structure.**

Below, we address the reviewers’ common questions.

## **Common Issue #1. Major Contribution** ##
Reviewers WXPj and fQfp expressed confusion about our major contribution. We are happy to clarify our contributions by providing a clear roadmap.

Our core contribution is innovating the existing 3DGS framework by leveraging mutual information, thereby enhancing the performance of various 3D editing tasks. Previously, JacobiNeRF introduced point-wise mutual information shaping based on NeRF for label propagation, which only allows one-time perturbation consistency. Extending JacobiNeRF's method to 3DGS for editing is nontrivial, because:

1. NeRF and 3DGS are fundamentally different types of representations. NeRF is a continuous scene representation modeled by a single MLP, whereas 3DGS represents a scene as a discrete set of millions of independent 3D Gaussians.To address the challenge of associating these independent Gaussians, we introduce a shared attribute decoding network that decodes the attributes for all Gaussians. By applying a similar mutual information shaping technique as in JacobiNeRF to this updated 3DGS framework, **we establish a new baseline for correlation shaping in 3DGS, named JacobiGS**. For a fair comparison, **we maintain consistency in other loss terms and supervision refinement (video tracker and regularization loss) within JacobiGS, as in our proposed method.**

2. To achieve 3D editing beyond label propagation, we modify JacobiNeRF’s shaping loss based on further derivations of the Jacobians. Our detailed theoretical analysis, provided in Appendix C, leads to the final result in Eq. 7. This new shaping loss ensures correlation consistency among Gaussians after parameter adjustments, without the need for the direct calculation of higher-order gradients in JacobiNeRF. **By solely replacing the original shaping loss in the JacobiGS framework with the new shaping loss, our method supports versatile and consecutive 3D editing**. In the task of 3D object movement (as shown in Fig. 6), existing techniques fail after three steps of continuous scene manipulation. In contrast, our method successfully maintains consistency throughout continuous editing, affecting only the target object. We have included videos from different views and scenes in the supplementary materials to persuasively illustrate the differences between JacobiGS and our method in object movement tasks involving successive operations. For tasks like 3D segmentation that do not require consecutive manipulations, as demonstrated in Table 1, JacobiGS performs worse compared to existing 3D segmentation methods. It indicates our proposed loss not only ensures consistency but also performs better in shaping correlations among Gaussians.
3. Furthermore, the original JacobiNeRF method is computationally expensive. As shown in Table 1, JacobiGS takes almost ten times longer for fine-tuning (excluding the time required for reconstruction, which is the same). The batch size for JacobiGS per epoch is only 48 (ours uses 512), and we maintain the same number of fine-tuning iterations for both JacobiGS and our method (refer to “training hyperparameters” in Appendix A). Additionally, the Jacobian calculation becomes extremely computationally intensive when the network layer being optimized is far from the network output. The computation path of Jacobian is longer in this case. **In practice, optimizing the Jacobian of the second layer and sampling only 128 Gaussians per epoch for contrastive learning in JacobiGS exceeds the memory limits of a single A100 GPU and requires hours for fine-tuning.**

In conclusion, our motivation is fundamentally different from JacobiNeRF, which focuses on label propagation. Our contribution and novelty lie in providing a theoretical analysis for consistent mutual information shaping, along with proposing a training pipeline with lightweight optimization that yields superior performance in 3D editing tasks.

---

> ### Author Response · Authors · 2024-11-20
> **Response addressing all Reviewers (Part 2)**
>
> ## **Common Issue #2. Effectiveness of mask guidance and regularization** ##
> Reviewers WXPj and fQfp raised questions about the effectiveness of mask guidance (video tracker and coarse mask labeling) and our regularization. We provide ablation studies for both quantitative and qualitative experiments. As shown in the last two rows of Table 1, the video tracker significantly improves segmentation quality. However, even without the associated masks produced by the tracker and regularization, our shaping technique still outperforms previous methods. This confirms that the performance improvement is primarily due to our proposed correlation shaping. Additionally, we provide a quantitative analysis of the regularization loss in Figure 13, Appendix B, which demonstrates that regularization further enhances correlation shaping, thus reducing artifacts generated after editing.
>
> Regarding the comparison with Gaussian Grouping mentioned by Reviewer WXPj, Gaussian Grouping employs the same 2D video mask tracker and a similar regularization loss. For a fair comparison, we directly use the associated masks provided by Gaussian Grouping. Thus, the observed performance improvement is attributable to our proposed shaping technique. Even without the video tracker and regularization loss, our method still outperforms Gaussian Grouping.
>
>
> ## **Common Issue #3. More experimental results** ##
> Reviewers fQfp and fawD requested additional experimental results, which we have provided in the appendix and supplementary materials. Videos of object movement showing consecutive perturbations are included in the supplementary materials, covering diverse views and scenes. The results of style transfer are presented in Appendix B. Comparisons with JacobiNeRF can be found in Table 1 of the revised paper. We hope these results demonstrate the credibility of our proposed methods, and we are happy to provide more if needed.
>
> We thank the reviewers for pointing out the concerns. **More detailed explanations for each reviewer could be seen in the personal response**. We hope that the information and clarifications provided in this rebuttal address your concerns and help you in re-evaluating our work.

---

### Meta-Review · Area_Chair_PYTd · 2024-12-19

**Metareview:**

This paper introduces a method using mutual information shaping technique for 3D Gaussian splatting. It has the advantages of efficient scene editing through attribute decoding networks, rather than direct manipulation of individual Gaussians. While the method inspired by JacobiNeRF, it offers significant contributions by applying the approach to 3D Gaussian fields, and achieving robust performance in open-vocabulary segmentation and editing tasks. Through concerns about limited technical novelty and weaker demonstrations, after the discussions, reviewers agreed on the innovative application of mutual information shaping, and the qualitative improvements in scene editing. I’d recommend the authors to consider all comments in the revision.

**Additional Comments On Reviewer Discussion:**

There were several rounds of discussion addressing concerns, such as novelty/contribution, effectiveness of mask guidance, and validation of the results . These concerns were partially resolved and acknowledged by the reviewers after discussion.

---

### Decision · Program_Chairs · 2025-01-22

Accept (Poster)